# Molecular Docking Assessment of Cathinones as 5-HT_2A_R Ligands: Developing of Predictive Structure-Based Bioactive Conformations and Three-Dimensional Structure-Activity Relationships Models for Future Recognition of Abuse Drugs

**DOI:** 10.3390/molecules28176236

**Published:** 2023-08-24

**Authors:** Nevena Tomašević, Maja Vujović, Emilija Kostić, Venkatesan Ragavendran, Biljana Arsić, Sanja Lj. Matić, Mijat Božović, Rossella Fioravanti, Eleonora Proia, Rino Ragno, Milan Mladenović

**Affiliations:** 1Kragujevac Center for Computational Biochemistry, Department of Chemistry, Faculty of Science, University of Kragujevac, Radoja Domanovića 12, P.O. Box 60, 34000 Kragujevac, Serbia; 2Department of Pharmacy, Faculty of Medicine, University of Niš, Bulevar Dr. Zorana Đinđića 81, 18000 Niš, Serbia; majavujovic1@gmail.com (M.V.); emilija293@gmail.com (E.K.); 3Department of Physics, Sri Chandrasekharendra Saraswathi Viswa Mahavidyalaya, Kanchipuram 631561, Tamil Nadu, India; ragav910@gmail.com; 4Faculty of Sciences and Mathematics, University of Niš, Višegradska 33, 18000 Niš, Serbia; ba432@ymail.com; 5Department of Science, Institute for Informational Technologies, University of Kragujevac, Jovana Cvijića bb, 34000 Kragujevac, Serbia; sanjamatic@kg.ac.rs; 6Faculty of Science and Mathematics, University of Montenegro, Džordža Vašingtona bb, 81000 Podgorica, Montenegro; mijatboz@ucg.ac.me; 7Department of Drug Chemistry and Technology, Faculty of Pharmacy and Medicine, Rome Sapienza University, P.le A. Moro 5, 00185 Rome, Italy; rossella.fioravanti@uniroma1.it; 8Rome Center for Molecular Design, Department of Drug Chemistry and Technology, Faculty of Pharmacy and Medicine, Rome Sapienza University, P.le A. Moro 5, 00185 Rome, Italy; eleonora.proia@uniroma1.it (E.P.); rino.ragno@uniroma1.it (R.R.)

**Keywords:** cathinones, 5-HT_2A_R, molecular docking, 3-D QSAR

## Abstract

Commercially available cathinones are drugs of long-term abuse drugs whose pharmacology is fairly well understood. While their psychedelic effects are associated with 5-HT_2A_R, the enclosed study summarizes efforts to shed light on the pharmacodynamic profiles, not yet known at the receptor level, using molecular docking and three-dimensional quantitative structure–activity relationship (3-D QSAR) studies. The bioactive conformations of cathinones were modeled by AutoDock Vina and were used to build structure-based (SB) 3-D QSAR models using the Open3DQSAR engine. Graphical inspection of the results led to the depiction of a 3-D structure analysis-activity relationship (SAR) scheme that could be used as a guideline for molecular determinants by which any untested cathinone molecule can be predicted as a potential 5-HT_2A_R binder prior to experimental evaluation. The obtained models, which showed a good agreement with the chemical properties of co-crystallized 5-HT_2A_R ligands, proved to be valuable for future virtual screening campaigns to recognize unused cathinones and similar compounds, such as 5-HT_2A_R ligands, minimizing both time and financial resources for the characterization of their psychedelic effects.

## 1. Introduction

The aminergic family of G-protein coupled receptors (GPCRs) physiologically respond to hormonal and neurotransmitter stimuli by activating internal signal transduction pathways and cellular responses [1], thus representing a valuable target class for drug discovery [2]. The 5-HT receptors (5-HTRs), as the largest subfamily of GPCRs, are activated by the neurotransmitter and neuromodulator serotonin (i.e., 5-hydroxytryptamine, 5-HT). After being biosynthesized in the intestine from tryptophan, the hormonal behavior of 5-HT in the periphery is associated with mitogenesis and the proliferation of fibroblast cells [3], homeostasis of glucose and lipid metabolism [4], enhanced arterial contraction during cardiac hypertrophy [5,6], bronchoconstriction [7], rhythmic breathing, and circadian rhythms [8], as well as pain, increased appetite, body temperature, etc. [9]. Nevertheless, small amounts of serotonin are biosynthesized in the axons of brain neurons, where 5-HT acts within the CNS (i.e., neocortex and hippocampi of the brain) as a major site for the expression of 5-HT receptors (5-HTRs), regulating mood, social cognition [10], neurogenesis [11], short- and long-term memory [12], spatial memory that enables navigation, sexual behavior, impulsivity, aggression, and migraine attacks [13,14].

At least 14 different serotonin receptors (5-HTRs), divided into seven subgroups (i.e., 5-HT_1-7_R), constitute the serotonergic system, primarily located on the membranes of either presynaptic or postsynaptic neurons [15]. The studied 5-HT_2A_R here (Figure 1), together with the 5-HT_2B_R and 5-HT_2C_R, is expressed on the postsynaptic membrane and acts as a mediator between the extracellular physiological ligand serotonin and intracellular G-proteins [1,16]. Activation of the 5-HT_2A_R by 5-HT induces the involvement of the Gα_i_, Gα_q/11_, or Gα_s_ proteins and subsequently increases the cellular levels of inositol triphosphate (IP3) and diacylglycerol (DAG) [17]. The 5-HT_2A_R-mediated pharmacology distinguishes the macromolecule as a drug target in the treatment of Parkinson’s and Alzheimer’s diseases, as well as mental disorders such as schizophrenia, bipolar disorder, depression, anxiety, and insomnia [16]. On the other hand, the 5-HT_2A_R is also a target for a variety of recreational drugs, mediating the effects of potent psychoactive substances such as lysergic acid diethylamide (LSD) or amphetamines, commonly known as psychedelics or hallucinogens [1,18].

Although a significant number of ligands co-crystallized with 5-HT_2A_R were available and deposited in the Protein Data Bank at the start of this investigation (Table 1, 13 ligands found in 14 complexes), only nine of them were homogeneously associated with inhibition constants (i.e., *K*_i_s) and thus were insufficient for the development of broad and computationally applicable medicinal chemistry models. Therefore, attention was focused on the literature concerning amphetamines and their β-keto analogs cathinones, as compounds known to exert 5-HT_2A_R-mediated psychedelic effects and well characterized by corresponding inhibition constants (Table 2), to elucidate their hitherto unknown binding modes and to describe their pharmacodynamic properties using 3D QSAR models. The 5-HT_2A_R ligands (besides those found co-crystallized within 5-HT_2B_R, Table 1) were further used either for defying structure-based alignment assessment (SBAA) rules, which were further employed for the generation of bioactive conformations of cathinones, or as an ultimate prediction test set (TS_CRY_) for external validation of 3-D QSAR models (see further discussion).

As natural products of the khat plant (*Catha edulis (Vahl) Forssk. ex Endl.*), cathinones were initially structurally optimized for the medical treatment of parkinsonism, obesity, and depression, but are now used as “legal highs” (also known as “bath salts”, “research chemicals not for human consumption”, or “plant food”) [19]. Recently developed DFT-based protocols have successfully reproduced experimental spectral data of selected synthetic cathinones (**SCs**) and can be further used for the identification/prediction of physicochemical parameters of either known or new **SCs**, even in forensic applications [20]. **SCs** primarily target presynaptic plasma membrane transporters for dopamine, norepinephrine, and serotonin (DAT, NAT, and SERT, respectively) [21]. Nevertheless, they increase the concentration of 5-HT in the synaptic space, providing the basis for CNS stimulatory and sympathomimetic effects characterized by increased blood pressure, heart rate, mydriasis, and hyperthermia [22]. As monoamine releasers, they enhance neurotransmitter transmission and increase synaptic concentrations; as monoamine reuptake inhibitors, they prevent the return of neurotransmitters to presynaptic neurons and consequent metabolic degradation while simultaneously activating monoamine receptors within the limbic-corticostriatal pathway [23].

The signaling and neurobehavioral effects of hallucinogens are associated with a 5-HT_2A_R expression on cortical layer V pyramidal neurons [17,24], suggesting the investigation of **SCs** as potential ligands for this receptor. Through 5-HT_2A_R binding, **SCs** can induce euphoria, increased empathy, sociability, energy, and alertness, but also rhabdomyolysis and autonomic symptoms such as tachycardia, hypertension, hyperthermia, ecstasy, sociability, alertness, empathy, and increased energy [21,22]. However, when taken at higher doses or for longer periods, stimulants can cause many psychiatric symptoms, including anxiety, agitation, and fear, which can progress to psychosis (delusions and hallucinations) and delirium [19,22]. Compared with amphetamines, several **SCs**, such as α-PPP (**9**) and MDPV (**14**) (Table 1), have more severe psychiatric side effects [21]. 

Since the mechanisms by which **SCs** exert the above-described effects as 5-HT_2A_R ligands are not yet fully understood and defined, the present study investigated the putative bioactive conformations and structure-based (SB) pharmacodynamic profiles of **SCs** (Table 2). The binding modes of the **SCs**, explored by molecular docking and 3-D QSAR studies were associated with their reported pharmacological profiles, providing a basis for the elucidation of the psychedelic or hallucinogenic effects of the **SCs** at the molecular level. Derived SB 3-D structure–activity relationships were found in good agreement with various ligands, either reported in the literature or experimentally resolved as 5-HT_2A_R ligands, for which the obtained results can be used in future virtual screening campaigns to reveal new **SCs** and similar compounds not yet recognized as 5-HT_2A_R ligands, minimizing both time and financial resources for the characterization of their potential psychedelic effects. 

## 2. Results and Discussion

### 2.1. Crystal Dataset Compilation 

The experimentally resolved complexes of ligands co-crystallized with either 5-HT_2A_R or 5-HT_2B_R (Table 1) were retrieved from the Protein Data Bank (PDB, https://www.rcsb.org/, accessed on 1 December 2022). Fourteen experimental 5-HT_2A_R/ligand complexes were assembled, including the full 5-HT_2A_R antagonists (FAs) and antipsychotics risperidone (PDB ID: **6A93**) [16], and zotepine (PDB ID: **6A94** [16]), a 5-HT_2A_R agonist (AG) and psychotic LSD (PDB IDs: **6WGT** [25] and **7WC6** [26]), the inverse agonist (IA) 1-methyl-4-[(5~{S})-3-methylsulfanyl-5,6-dihydrobenzo[b][1]benzothiepin-5-yl]piperazine (PDB ID: **6WH4** [25]), the 5-HT_2A_R partial antagonist (PA) and hallucinogen 25-CN-NBOH (PDB ID: **6WHA** [25]), the 5-HT_2A_R agonist (3*R*)-3-methyl-5-(1*H*-pyrrolo[2,3-b]pyridin-3-yl)-1,2,3,6-tetrahydropyridin-1-ium (**7RAN** [27]), the FAs cariprazine (PDB ID: **7VOD** [28]) and aripiprazole (PDB ID: **7VOE** [28]), the AGs serotonin (PDB IDs: **7WC4** [26])psilocin (the active metabolite of psilocybin, PDB ID: **7WC5** [26]), lisuride (PDB ID: **7WC7** [26]), and FAs lumateperone (PDB ID: **7WC8** [26]), and the non-hallucinogenic psychedelic analog IHCH-7113 (PDB ID: **7WC9** [26]).

The 5-HT_2B_R ligands were AGs ergotamine (PDB IDs: **4IB4** [29], **4NC3** [30] and **5TUD** [31]), LSD (PDB IDs: **5TVN** [31], **7SRR** [32] and **7SRS** [32]), the agonists *N*,*N*-diethyl-*N*’-[(8α)-6-methyl-9,10-didehydroergolin-8-yl]urea (PDB ID: **6DRX** [33]), (8β)-*N*-[(2*S*)-1-hydroxybutan-2-yl]-6-methyl-9,10-didehydroergoline-8-carboxamide (PDB ID: **6DRY** [33]), (8α)-*N*-[(2S)-1-hydroxybutan-2-yl]-1,6-dimethyl-9,10-didehydroergoline-8-carboxamide (PDB ID: **6RDZ** [33]), as well as FA (1*S*)-1-[(2-chloro-3,4-dimethoxyphenyl)methyl]-6-methyl-2,3,4,9-tetrahydro-1*H*-beta-carboline (PDB ID: **6DS0** [33]).

### 2.2. Literature Datasets Set Compilation

The **SCs** as 5-HT_2A_R ligands were retrieved from the literature to compile a training set (TR) (Table 2) [21,34,35,36,37,38,39,40,41,42,43,44,45], whereas the test set (TS) was compiled from listed clinically approved drugs (Table 3) [46,47,48,49,50,51,52,53,54,55,56,57,58,59,60,61,62] with known affinities for 5-HT_2A_R, but with unknown binding modes for all compounds except **25** (found within the **7VOE** protein [28]). For either TR or TS compounds, analysis of the available biological data revealed a homogeneous association with potencies described as p*K*_i_s (-log*K*_i_). Thus, the TR included **Cathinone** (1) and its derivatives, **Flephedrone** (**2**), **Mephedrone** (**3**) (the most commonly abused cathinone, non-selective monoamine uptake inhibitor [34,35]), and **Methcathinone** (**4**), as likely AGs and psychedelic compounds, as well as FAs and rather antidepressive drugs **4-Bromomethcathinone** (**5**), **3-Bromomethcathinone** (**6**, [21]), **2-Fluoromethcathinone** (**7**), and **2-Trifluoromethoxy-methcathinone** (**8**), also known to act as preferential dopamine active transporter (DAT) and noradrenaline transporter (NAT) inhibitors and α_1A_ adrenoceptor agonists/antagonists [21,34].

The experimentally determined 5-HT_2A_R IAs/FAs and non-psychostimulants pyrovalerone-based **SCs**, namely **α-PPP** (**9** [21]), **4-Methyl-α-PPP** (**10** [21]), **4-Bromo-α-PPP** (**11**), and **3-Bromo-α-PPP** (**12**), were also included in the TR, alongside with **Naphyrone** (**13**), **MDPV** (**14**), **Pyrovalerone** (**15**), and **MDPPP** (**16**) as experimentally validated FAs, also known as norepinephrine and dopamine reuptake inhibitors (NRIs and DRIs, respectively) [36,37].

**Table 1 molecules-28-06236-t001:** Names, structures, and inhibition constants of co-crystalized 5-HT_2A_R and 5-HT_2B_R ligands.

PDB IDP (Mechanism) ^a^	Compound’sStructure	p*K*_i_	Ref.	PDB IDP (Mechanism) ^a^	Compound’sStructure	p*K*_i_	Ref.
*5-HT_2A_R ligand*	**7WC7**AG	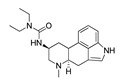	8.55	[26]
**6A93**FA ^a^	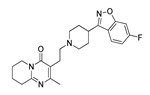	8.16	[16]	**7WC8**AG	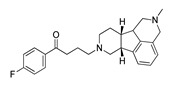	9.27	[26]
**6A94**FA	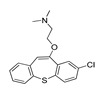	7.40	[16]	**7WC9**AG	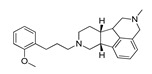	NA	[26]
**6WGT****7WC6**AG	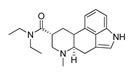	8.63	[16][26]	*5-HT_2B_R ligand*
**6WH4**IA	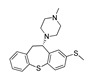	9.70	[25]	**4NC3****4IB4****5TUD**AG	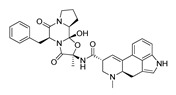	8.88	[29][29][31]
**6WHA**PA	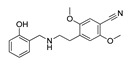	9.08	[25]	**5TVN**AG	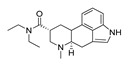	NA	[31]
**7RAN**AG	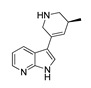	NA ^a^	[27]	**6DRX**AG	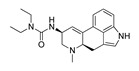	8.88	[33]
**7VOD**FA	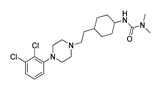	7.73	[28]	**6DRY****7SRQ****7SRS****7SRR**AG	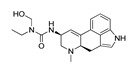	9.33	[33][32][32][32]
**7VOE**FA	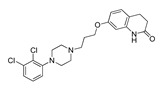	8.47	[28]	**6DRZ**AG	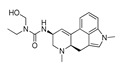	10	[33]
**7WC4**AG	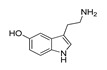	7.49	[26]	**6DS0** **FA**	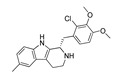	NA	[33]
**7WC5**AGPA	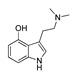	NA	[26]				

^a^ Pharmacology: AG—agonist, IA—inverse agonist, PA—partial agonist, FA—full antagonist

**Table 2 molecules-28-06236-t002:** Names, structures, and inhibition constants of cathinones as human 5-HT_2A_R ligands compiling the TR.

Name(Number) P (Mechanism) ^a^	Compound’sStructure	p*K*_i_	Ref.	Name(Number) P (Mechanism) ^a^	Compound’sStructure	p*K*_i_	Ref.
*Cathinone and its derivatives*	**Naphyrone****(13)** FA	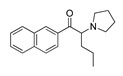	4.96	[34]
**Cathinone****(1)** AG ^a^	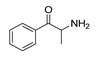	6.00	[34]	**MDPV****(14)** FA	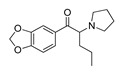	4.88	[34]
**Flephedrone****(2)** AG	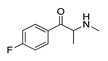	6.00	[34]	**Pyrovalerone****(15)** FA	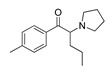	4.88	[34]
**Mephedrone****(3)** AG	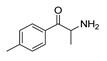	5.68	[34]	**MDPPP****(16)** FA	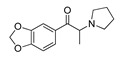	4.20	[21]
**Methcathinone****(4)** AG	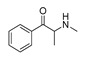	5.23	[34]	*Benzo[d][1,3]dioxole-based **SCs***
**4-Bromomethcathinone****(5)** IA/FA	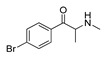	5.20	[21]	**MDBD****(17)** AG	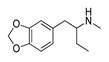	5.20	[34]
**3-Bromomethcathinone****(6)** IA/FA	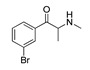	5.00	[21]	**MDMA****(18)** AG	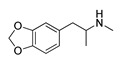	5.11	[34]
**2-Fluoromethcathinone****(7)** IA/FA	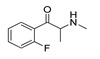	5.00	[21]	**Butylone****(19)** AG	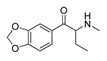	4.88	[34]
**2-(Trifluoromethoxy)****-methcathinone****(8)** IA/FA	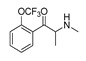	5.00	[21]	**Ethylone****(20)** AG	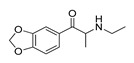	4.88	[34]
*Pyrovalerone-based **SCs***	**MDEA****(21)** AG	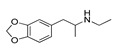	4.88	**MDEA** **AG (21)**
**α-PPP****(9)** IA/FA	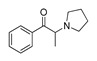	5.60	[21]	**Methylone****(22)** AG	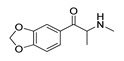	4.88	[34]
**4-Methyl-α-PPP****(10)** IA/FA	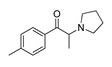	5.50	[21]	** *SCs* ** *’ precursors*
**4-Bromo-α-PPP****(11)** IA/FA	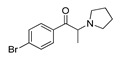	5.40	[21]	**Amphetamine****(23)** AG	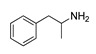	4.88	[34]
**3-Bromo-α-PPP****(12)** IA/FA	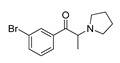	5.00	[21]	**Methamphetamine****(24)** AG	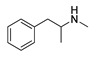	4.88	[34]

^a^ Pharmacology: AG—agonist, IA—inverse agonist, PA—partial agonist, FA—full antagonist.

**Table 3 molecules-28-06236-t003:** Commercial names, structures, and inhibition constants of human 5-HT_2A_R ligands compiling the TS.

Name(Number) P (Mechanism) ^a^	Compound’sStructure	p*K*_i_	Ref.	Name(Number) P (Mechanism) ^a^	Compound’sStructure	p*K*_i_	Ref.
**Aripiprazole****(25)** FA ^a^	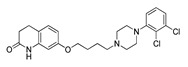	8.57	[46]	**Norfenfluramine****(36)** AG	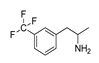	6.82	[47]
**BW-723C86****(26)** AG	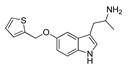	7.2	[48]	**Olanzapine****(37)** IA	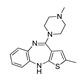	8.88	[49]
**Clozapine****(27)** FA	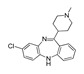	8.39	[50]	**Quentiapine****(38)** FA	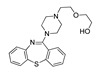	6.81	[50]
**CP-809,101****(28)** AG	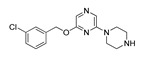	8.22	[51]	**R060-0175****(39)** IA	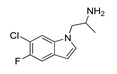	7.44	[48]
**Ketanserin****(29)** FA	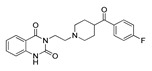	9.67	[52]	**Risperidone****(40)** IA	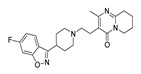	9.69	[50]
**Lorcaserin****(30)** AG	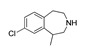	6.95	[53]	**RS-127,455****(41)** FA	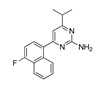	6.03	[48]
**MDL-100,907****(31)** FA	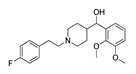	8.77	[54]	**Saprogrelate****(42)** FA	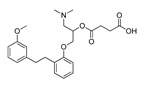	8.52	[55]
**Mesulergine****(32)** FA	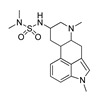	7.34	[56]	**SB-204,741****(43)** FA	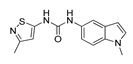	5.00	[57]
**Mianserin****(33)** FA	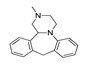	8.15	[58]	**SB-206,553****(44)** FA	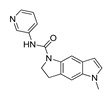	5.64	[59]
**Mirtazapine****(34)** IA	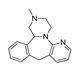	7.78	[60]	**SB-242,084****(45)** FA	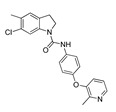	6.07	[61]
**Naftidrofuryl****(35)** IA	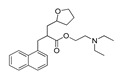	6.20	[55]	**WAY-161,503****(46)** AG	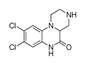	7.40	[62]

^a^ Pharmacology: AG—agonist, IA—inverse agonist, PA—partial agonist, FA—full antagonist.

Benzo[d][1,3]dioxole-based **SC**s, **MBDB** (**17**), **MDMA** (ecstasy, **18**), **Butylone** (**19**), **Ethylone** (**20**), **MDEA** (**21**), and **Methylone** (**22**) were also listed in the TRs as examples of **SCs** with the ability to directly bind 5-HT_2A_R most likely as AGs and psychostimulants, resulting in the release of 5-HT. Of these, **18** induces excitation and hallucinogenic-like perceptual changes at higher doses [38,39,40], while **19**, **20**, and **22** are known as non-selective monoamine uptake inhibitors [33,34,35,36]. The precursors of **SCs**, **amphetamine** (**23**) and **methamphetamine** (“crystal meth”, **24**), likewise considered to be AGs, completed the TR compilation. Binding to 5-HT_2A_R, **23** and **24** induce behavioral effects opposite to those induced by stimulation of 5-HT_1A_R (5-HT_2A_R-mediated depolarization vs. the 5-HT_1A_R-mediated hyperpolarization) [18,41].

Since the quantitative parameters for psychedelic or hallucinogenic effects, such as the onset of action (OA) and duration of action (DA), are known for only a few compounds (**1**–**8** [35,43,44,45], **13**, **14**, and **16** [36,37], and **23** and **24** [34], Appendix A), it is not possible to derive a robust quantitative psychoactivity relationship model. Therefore, the pK_i_s of the **SCs** (Table 2) were correlated with the SB bioactive conformations of their 5-HT_2A_Rs using 3-D QSAR models, whose predictive abilities were externally validated by the compiled TS.

### 2.3. Definition of Optimal Protocol for the Alignment of 5-HT_2A_R Ligands

In the development of the CoMFA-like 3-D QSAR models, a crucial step is represented by the alignment rules [63]. Therefore, considering the lack of **SCs** co-crystallized with 5-HT_2A_R (Table 2), their bioactive conformations were modeled using as much experimentally available structural data as possible on 5-HT_2A_Rs and 5-HT_2B_Rs crystallized in complexes with different ligands (Table 1) [16,25,26,27,28,29,30,31,32,33,64]. Attention was initially focused on lysergic acid diethylamide (LSD), the prototypical human hallucinogen, one of the most potent psychoactive (and recreational) drugs [31], and an experimentally confirmed agonist for both 5-HT_2A_R (PDB IDs: **6WGT** and **7WC6**, [25,26] and 5-HT_2B_R (PDB IDs: **5TVN** [31], **6DRX** [33], **7SRR** [32], and **7SRS** [32]) isoforms [64].

Regardless of the active site, LSD showed an almost constant binding conformation with the lowest root mean square deviation (RMSD) (Table 4). Sequence and active site similarities of 5-HT_2A_R and 5-HT_2B_R were approximately 40% and 60%, respectively (Figure 2), therefore both targets were used to establish a protocol for the alignment of **SCs** in the 5-HT_2A_R. To this end, an alignment evaluation of ligands co-crystallized in either 5-HT_2A_R (Figure 3a, Table 4) [16,25,26,27,28] or 5-HT_2B_R (Figure 3b, Table 5) [29,30,32,33] was performed to establish the alignment rules to define the binding modes of TR and TS compounds. The quality of alignment was evaluated by SB 3-D QSAR models in terms of *q*^2^ value [65].

#### Structure-Based Alignment Assessment

Either ligands/5-HT_2A_R (Table 5) or ligands/5-HT_2B_R (Table 6) experimentally resolved complexes were superimposed using **7WC8** [26,66] as a template, with the lowest resolution of 2.45 Å. For the SB alignment assessments (SBAAs) procedure [67], the best performing molecular docking algorithm/scoring function pair in reproducing the experimentally bound conformations of the ligands was investigated using ligands extracted from either 5-HT_2A_R (Table 5) or 5-HT_2B_R (Table 6) complexes (at the same time providing a basis for future cross-pharmacology in a discrete fashion) [67]. The following protocol was established: first, any ligand was subjected to SBAA [67] using free or open-source molecular docking programs: AutoDock [68], AutoDock Vina (hereafter Vina) [69], DOCK [70], SMINA [71], and PLANTS [72], with all available scoring function (SF) variants. As previously described, the SBAA protocol [67] was investigated at four levels of difficulty: experimental or randomized ligand conformation (EC and RC, respectively), re-docking (RD) (ECRD and RCRD, respectively), and cross-docking (CD) (ECCD and RCCD, respectively). While the RMSD data associated with ECRD, RCRD, and ECCD are reported as Appendix A, the RCCD-related RMSD values, which describe the docking programs’ abilities in reproducing experimental binding modes from initial random ligand conformations into a never-seen protein environment (the most difficult scenario), are reported for either 5-HT_2A_R or 5-HT_2B_R as Table 5 and Table 6, respectively.

Regarding the reproduction of co-crystallized ligand conformations in 5-HT_2A_R (Table 5, Figure 3a), in the ECRD phase (Appendix A), all available docking algorithms/SFs pairs, except those implemented in AutoDock and DOCK, showed a high level of docking accuracy (DA) [67,73,74]. Among the most accurate, Vina was found to be the best, with a DA value of 89.28%, while SMINA, with either vina [69], vinardo [75], or ad4 [68] SFs, showed DA ranging from 75.00% (SMINA/vina) to 78.57% (SMINA/vinardo). The PLANTS/plp pair (DA = 82.14%) almost reached the accuracy of Vina, while either PLANTS/chemplp or PLANTS/lp95 gave DA values close to SMINA/vinardo. At the RCRD stage (Appendix A), Vina maintained a DA value of more than 80%, with a physiological decrease of about 7% compared to ECRD, while each SMINA/SF pair-associated DA was less than 75%. The ability of PLANTS/plp to reposition 5-HT_2A_R’ ligand conformations showed a 5% decrease in DA, while PLANTS/chemplp remained as accurate as SMINA/vinardo. In the ECCD evaluation phase (Appendix A), Vina remained the most accurate (DA equal to 75%), followed by SMINA/vina and SMINA/vinardo (DA ~ 72%), while the SMINA/ad4 and PLANTS pairs, plp and plp95, were 50–60% accurate. At the last and most difficult level, the RCCD stage (Table 5), Vina proved to be the docking program with the highest DA (50%).

As for the docking evaluation for ligands co-crystallized in 5-HT_2B_R (Table 6, Figure 3b), all docking algorithms/SF pairs showed DAs higher than 50% during the ECRD phase (Appendix A). In particular, Vina and SMINA/vina/vinardo showed perfect DAs (100%), followed by SMINA/ad4 (DA higher than 80%), while for AutoDock, DOCK, and PLANTS, the DAs ranged between approximately 59% (DOCK) and 73% (PLANTS/chemplp/plp). In the RCRD assessment (Appendix A), Vina retained the highest DA (86.36%), while SMINA was associated with DAs of approximately 68% and 73%. A decreasing trend was also observed for AutoDock, DOCK, and PLANTS (chemplp and plp SFs) with DA values equal to or greater than 50% but not exceeding 65%. At the ECCD level (Appendix A), Vina and SMINA/vina/vinardo/ad4 were the only ones able to cross-dock 5-HT_2B_R ligands with acceptable DA values of 72.73, 63.64, 54.00, and 50.00%, respectively, while AutoDock, DOCK, and all PLANTS SFs suffered from the initial conformation with low DA values. Finally, during the RCCD (Table 4), as in the case of 5-HT_2A_R, Vina was confirmed as the program that was able to reproduce the experimental binding conformations with the least error (DA = 63.64%).

### 2.4. **SCs**’ Binding Mode Analysis into 5-HT_2A_-DPPC and 3-D QSAR Models Interpretation

Based on the above SB assessments, Vina was selected as an SB tool to investigate the binding modes of either TR or TS molecules. SCs were cross-docked into the **6A93** [16] crystal of 5-HT_2A_R, previously immersed in the dipalmitoylphosphatidylcholine (DPPC) membrane system (5-HT_2A_R-DPPC, https://opm.phar.umich.edu/, entry: 4282, accessed on 2 December 2022), and co-aligned with the 5-HT_2A_R and 5-HT_2B_R crystals used for SBAAs (Table 3 and Table 4) to elaborate their binding modes, simulating a cell-like environment. TR compounds thus adopted a binding mode similar to that of LSD and were found in the vicinity of extracellular loop 2 (EL2) and the extracellular space, which forms a narrow cleft lined mainly by hydrophobic side chains of residues in transmembrane (TM) helices TM3, TM5, TM6, and TM7, as well as near the laterally extended cavity surrounded by hydrophobic residues of TM3, TM4, TM5, and ECL2, to connect the binding site to the plasma membrane near the lower hydrophobic cleft [16]. None of the compounds occupied the orthosteric region in front of the hydrophobic cleft [16]. Upon deeper analysis, **SCs** were observed to establish only electrostatic interactions with TM3 Asp155, instead of a hydrogen bond (HB) established by LSD, which has been identified as a crucial interaction for the drug’s psychotic behavior at the 5-HT_2A_R level [76]. To quantitatively support the observed SB overlap, 3-D QSAR models were built using the TR docked conformations proposed by Vina. The 3-D QSAR models were generated using Open3DQSAR software [77] with multiple probes (Table 7), and the corresponding CoMFA-like maps were analyzed. The final models were obtained with the fractional factorial design (FFD) feature selection applied to the model preoptimized with the variable pretreatment optimization (VPO) protocols by varying the lattice spacing and lattice extension [78].

The best models were obtained with the OH2 probe, with associated q^2^ values of 0.684, 0.562, and 0.671 with steric (STE), electrostatic (ELE), and both fields (BOTH or STE + ELE), respectively (Table 7, Figure 4). Although the OH2 probe STE model was statistically superior to the model derived with either field (BOTH), for the sake of completeness, the final model interpretation was performed using the OH2 probe-based PLS-coefficients (OH2*_PLS-coefficients_* maps) obtained with the BOTH field model. Thus, in a CoMFA-like fashion, the positive STE OH2*_PLS__-coefficients_* (green isocontours) cover the regions of the molecules where the bulky substituents would increase the activity, while the negative STE OH2_PLS-_coefficients (shown as yellow isopleths) indicate the regions where the bulky substituents decrease the activity. Positive ELE OH2*_PLS__-coefficients_* (red polyhedra) indicate regions where positively charged functions and hydrogen bond donors (HBDs) are positively correlated with activity, while negative ELE OH2*_PLS__-coefficients_* (blue maps) indicate regions where negatively charged functions and hydrogen bond acceptors (HBAs) are positively correlated with potency. Despite the intrinsic limitations of the generated OH2-based STE, ELE, and BOTH models (limited TR chemical diversity), the low error of the cross-validated absolute error of prediction for either TR or TS and TS_CRY_ activities (Appendix A) enabled the model as a tool to predict the activities of yet-untested SCs analogs or similar compounds (see further discussion).

#### 2.4.1. The Cathinones’ Benzene Ring and Its Substituents’ Contribution

The benzene ring (**Ph**) of the **SCs** can be considered as a first pharmacophoric feature. Thus, the unsubstituted **Ph** (found in AGs **1**, Figure 5a,b; **4**, Figure 6a,b; **23**, Appendix A, and **24**, Appendix A, as well as within an IA **9**, Figure 6c,d; Appendix A) was placed at the bottom of the hydrophobic cleft. Within the listed **SCs**, the **Ph**’s *ortho*-positions proximal to the β-carbonyl group (**β-CO**) orientation (see further discussion), according to the positive STE OH2*_PLS-coefficients_* maps, formed favorable steric interactions with Ser159, Thr160, and Ser242 side chains, implying that the additional bulkiness would be preferable for the agonistic pharmacology and likely the hallucinogenic effects. In contrast, the yellow STE OH2*_PLS-coefficients_* maps emphasized the unfavorable T-shaped (i.e., edge-to-face) van der Waals interactions of the opposite-side *o*-positions with the benzyl part of Trp336′s indole ring, as well as with Phe340, toward which bulkiness should be reduced, perhaps endowing lower psychostimulation.

On the other hand, the *m*-positions on the **β-CO** side interacted with Ile163, where the negative steric 3-D QSAR maps indicated repulsive interactions with the *sec*-methyl group of Ile163, while the positive STE OH2*_PLS-coefficients_* maps indicated the favorable attraction with the pentanoic acid-based side chain of the residue, as well as with the indole ring of Trp164. The *m*-carbon atoms were observed in favorable T-shaped steric interactions with the indole ring nitrogen of Trp336 (the green STE OH2*_PLS-coefficients_* maps as evidence).

Finally, the *p*-positions could mainly establish van der Waals contacts with Phe332, Val333, and Phe340 in a T-shaped manner, as well as with Phe243 via parallel displaced interactions, described with either positive or negative steric 3-D QSAR maps. The slightly larger extent of the negative steric 3-D QSAR maps can probably be associated with a repulsive character.

In addition, upon the *p*-substitution of **Ph** with a halogen moiety (leading to the agonism of **2**, Figure 5c,d; full antagonism of **5**, Appendix A, and inverse agonism of **11**, Appendix AAppendix A), the *p*-halogen likely established induced dipole interactions with Phe332 and the surrounding residues, according to the negative ELE OH2*_PLS-coefficients_* maps. The slight decrease in electronegativity (and increase in bulkiness) upon *p*-Br incorporation, as in **11** or **5**, resulted in reduced inhibition constants and likely inverse agonism/full antagonism. Moreover, concerning **2**, both **11** and **5** were slightly rotated towards Trp336, overlapping the *m*-position of the unsubstituted benzene ring. Therefore, according to the positive ELE OH2*_PLS-coefficients_* fields, as with HBA, the *p*-Br was not well tolerated by the indole moiety of Trp336, which probably prefers an HBD at the *m*-position while interfering with **SC**; according to the alignment of the positive STE OH2*_PLS-coefficients_* contours, certain HBD could be of increased steric hindrance.

The *p*- to *m*-halogen position shift, i.e., *p*-Br to *m*-Br, as seen in **12** (Appendix A) or **6** (Appendix A), in which the *m*-Br pointed toward Phe332 and Trp336, likely contributed to the reduced binding potency, full antagonism, and the exertion of antidepressive effects. Nevertheless, the model also indicated that a double *m*-halogen substitution could contribute to an additional decrease in the binding potency of **SC**.

A decrease in potency is observed when the halogen position is changed from *para* to *ortho*, as seen in FAs and non-psychostimulants/antidepressives **7** (Appendix A) and similarly in **8** (Appendix A). The *o*-F moiety of **7** established unfavorable steric interactions (negative STE OH2*_PLS-coefficients_* contours) with the Trp336 side-chain benzene. At the same time, the positive ELE OH2*_PLS-coefficients_* isocontours indicated that an *o*-HDB within the structure of **SC** is preferred by Trp336 instead of *o*-F as HBA. The **8′**s *o*-OCF_3_ as a bulkier and more electronegative scaffold led to an unwanted steric clash (negative STE OH2*_PLS-coefficients_* contours as evidence) with Ile163 and Trp164, although the positive ELE OH2*_PLS-coefficients_*/negative ELE OH2*_PLS-coefficients_* isocontours alignment implied that more of the HBD character is needed for Trp164 engagement.

Replacement of the *p*-halogen with a hydrophobic moiety such as the methyl of an AG **3** (Appendix A), as well as IAs/FAs **10** (Appendix A) and **15** (Appendix A), resulted in a ~15° rotation of the central aromatic ring, leading to the overlap of the *p*-CH_3_ function with the *p*-Br or *m*-Br, respectively. The increased bulkiness at the bottom of the hydrophobic cleft reduced the potency, as verified by the incorporation of 1,3-dioxolane at the *m*- and *p*-carbons of the phenyl ring, respectively (i.e., benzo[d][1,3]dioxole formation), as in derivatives **14** (Appendix A) and **16**–**22** (Figure 5, Figure 6 and Appendix A), or fused benzene as the naphthyl moiety in **13** (Appendix A), contributing to inverse agonism/full antagonism (**14** and **15**).

#### 2.4.2. The Cathinones’ β-Carbonyl Group Contribution

The orientation of the unsubstituted or substituted **Ph** within the structures of **SCs** strongly influenced the behavior of **β-CO** and vice versa. This group can be considered as a second pharmacophoric feature, as in **1** (Figure 5a,b) and **4** (Figure 6a,b). The **β-CO** formed a strong HB with the side-chain hydroxyl portion of Thr160 (**β-CO**----HO-Thr160, O-O distance of 2.636 and 2.464 Å, respectively), characterized by negative ELE OH2_PLS*-coefficients*_ isocontours, thus contributing to the agonistic pharmacology and psychedelic effects. With the incorporation of *p*-F (**2**, Figure 5c,d), the **β-CO** has shifted away from preventing the optimal **β-CO**----HO-Thr160 O-O distance toward attractive electrostatic interactions (*p*-F likely compensated for the lack of HB to maintain the high potency of **2** as an AG and psychostimulant). However, with the introduction of the *p*-Br, as in **11** and **5** (Appendix A), the β-CO----HO-Thr160 hydrogen bonding distance was restored due to the ~15° in-plane rotation of **CSs** concerning **2** (O-O distance of 3.396 and 3.238 Å for 11 and 5, respectively), proving the value for IAs/FAs as well. The *p*-substitution with a methyl group (**3**, **10**, and **15**, Appendix A), together with both *p*- and *m*-substitution with 1,3-dioxolane or benzene (**13**, **14**, **16**–**22**, and Figure 5, Figure 6 and Appendix A), also forced the **β-CO**----HO-Thr160 HB formation (O-O distance of 2. 728, 3.360, 2.835, 3.321, 3.229, 2.989, 2.892, 3.119, and 3.217 Å, respectively).

However, the *p*-Br to *m*-Br switch, as in **12** (Appendix A) and **6** (Appendix A), or the *p*-F to *o*-F switch, as in **7** (Appendix A), all inverse agonists/full antagonists, resulted in the alternative **β-CO**----HO-Ser159 HB formation (O-O distances of 2.585 Å, 3.245 Å, and 3.396 Å, respectively), for which relatively high potency was retained. The **β-CO**----HO-Ser159 HB (O-O distance = 2.971 Å) was retained even when the bulkier **8′**s *o*-fluoromethoxy moiety (Appendix A) formed an additional HB with Thr160 (O-O distance = 2.288 Å), also covered by negative ELE OH2*_PLS-coefficients_* isocontours, or when **Ph** was unsubstituted (**9**, Figure 6c,d, *d*_HB_ = 2.971 Å). The absence of **β-CO** (**14** and **16**–**22**, and **23** and **24**, Figure 5 and Figure 6, and Appendix A) resulted in the lowest potencies, which were partially compensated by the benzo[d][1,3]dioxole (**14** and **16**–**22**) or **Ph** (**23**–**24**) alone.

#### 2.4.3. The Cathinones’ Methylene Group and Methylene Group’ Substituents Contribution

Here, we focused on the importance of alkyl-substituted methylene fragments (i.e., **-CH_2_**-Rs, either methyl- [**-CH_2_-**CH_3_], ethyl- [**-CH_2_-**CH_2_-CH_3_ ], or propyl- [**-CH_2_-**CH_2_-CH_2_-CH_3_] substituted) located between either substituted/non-substituted **Ph-β-CO** or benzyl (**Bn**, in the absence of **β-CO**) moieties and substituted/non-substituted terminal amines (-**NH_2_**) (Table 1). Therefore, the orientations of the **-CH_2_-**CH_3_ fragments were almost conserved and the correct SB correlation to the surrounding groups was difficult to establish. Thus, between either **Ph-β-CO**----HO-Thr160 HB and -**NH_2_** or -**NH**CH_3_ (**1**, Figure 4a and Figure 5b, and **4**, Figure 6a,b, respectively) or **Bn** and -**NH_2_** or -**NH**CH_3_ (**23**, Appendix A: **24**, Appendix A), the **-CH_2_-**CH_3_ was involved in the attractive steric interactions with Val156 (see the STE OH2*_PLS-coefficients_* maps) that enhanced the agonistic and hallucinogen effects of compounds. Nevertheless, the **-CH_2_-**CH_3_ contribution was not critical for the **SCs**, as it was the loss of **β-CO** that directly affected the potency reduction from the highest (**1**) to the lowest (**23** and **24**). The conversion of -**NH_2_** to pyrrolidine (**9**, Figure 6c,d) directed the substituent methyl group toward favorable interactions with Phe339 and Phe340 (as evidenced by positive STE OH2*_PLS-coefficients_* maps), but also contributed to inverse agonist/full antagonist pharmacology. Interestingly, a similar orientation of the **-CH_2_-**CH_3_ moiety (accompanied by a further decrease in potency) was observed with *m*-Br-**Ph-β-CO**----HO-Ser159 HB and pyrrolidine (**12**, Appendix A), as well as with *o*-OCF_3_-**Ph-β-CO**----HO-Ser159 HB and -**NH**CH_3_ (**8**, Appendix A). In addition, both **-CH_2_-**CH_3_ (**22**, Appendix A; **20**, Appendix A) and **-CH_2_-**CH_2_-CH_3_ (**19**, Appendix A) indicated that Phe339 and Phe340 were bound by 1,3-dioxolane-**Ph**-**β-CO**----Thr160 HB and either -**NH**CH_3_ or -**NH**-Et.

The potency remained relatively high (above the 5 p*K*_i_ units), with the **-CH_2_-**CH_3_ surrounded by *p*-X-**Ph-β-CO**----HO-Thr160 HB and either -**NH**CH_3_ (**2**, X=F, Figure 5c,d; **5**, X=Br, Appendix A) or pyrrolidine (**11**, X=Br, Appendix A), and by *p*-Me-**Ph-β-CO**----HO-Thr160 HB and pyrrolidine (**3**, Appendix A; **10**, Appendix A), which further directed the targeted functional group towards the T-shaped steric interaction with the indole ring of Trp336, where a larger portion of negative STE OH2*_PLS-coefficients_* contours relative to positive ones indicated that further bulkiness increase towards Trp336 would lead to inverse agonism/partial agonism and would thus be detrimental for the psychedelic influence of **SC**, whereas positive/negative ELE OH2*_PLS-coefficients_* maps indicated that methyl group replacement by HBD/HBA would be beneficial for agonistic behavior and psychedelic effects.

In contrast, either *m*-Br-**Ph-β-CO**----HO-Ser159 HB (**6**, Appendix A) or *o*-F-**Ph-β-CO** (**7**, Appendix A) next to -**NH**CH_3_ forced the **-CH_2_-**CH_3_ to have unproductive (note negative STE OH2*_PLS-coefficients_* contours) van der Waals interactions with Val156 and Ser242, which was reflected in inverse agonism/partial agonism pharmacology. The similar interactions of **-CH_2_-**CH_3_ were observed to be surrounded by 1,3-dioxolane-**Ph**-**β-CO**----Thr160 HB and pyrrolidine (**16**, Appendix A), as well as by 1,3-dioxolane-**Bn**-and-**NH**CH_3_ (**18**, Figure 6e,f; **21**, Appendix A), but not for **-CH_2_-**CH_2_-CH_3_ (**17**, Figure 5e,f), which remained bound to Phe340. However, the incorporation of the propyl group was accompanied by either naphthalene-**β-CO**----HO-Thr160 HB (**13**, Appendix A), 1,3-dioxolane-**Ph**-**β-CO**····HO-Thr160 HB (**14**, Appendix A), or *p*-Me-**Ph-β-CO**····HO-Thr160 HB (1**5**, Appendix A), and pyrrolidine turned the **-CH_2_-**CH_2_-CH_2_-CH_3_ towards Trp151 and Val156, defining the boundary for Phe340 steric clash tolerance.

#### 2.4.4. The Cathinones’ Amine Nitrogen Contribution

The affinities of **SCs** to 5-HT_2A_R and their psychedelic behavior can also be attributed to their interactions with Asp155 via the terminal primary, secondary, or tertiary nitrogen. Thus, the primary amine, as in AGs **1** (Figure 5a,b), **3** (Appendix A) and **23** (Appendix A), provided a salt bridge with Asp155 instead of HB (*d* = 4.772 and 5.853 Å, respectively, and the strongest interaction could have contributed to the highest potency of **1**), despite being characterized with positive ELE OH2*_PLS-coefficients_* maps. The absence of hydrogen bond between the -**NH_2_** and Asp155 is likely the reason for lower psychedelic activities of **SCs** compared to LSD [25,26,31,32]. Nevertheless, distinct interactions were characterized by negative STE OH2*_PLS-coefficients_* isocontours, indicating that the increase in bulkiness at the -**NH_2_** level would reduce the potency of **SCs** and convert AGs into IAs/FAs, i.e., psychostimulants into non-psychostimulants.

The above hypothesis was (in part) supported by the analysis of the secondary, i.e., either *N*-methyl substituted, as in **2**, **4**–**8**, **17**–**19**, **22**, and **24** (Figure 5, Figure 6 and Appendix A), or *N*-ethyl substituted, as in **20** and **21** (Appendix A) amines. The listed **SCs** also formed a salt bridge with Asp155 via the -*N*-H part (highlighted by positive ELE OH2*_PLS-coefficients_* isocontours), strongly conditioned by the alignment of the *N*-methyl/*N*-ethyl groups with one of the two subregions bounded by Leu228, Val336, Phe339, Trp336, and Tyr370 (as for **5**–**7** and **20**), or Val156 alone (as for **4**, **8**, **17**–**19**, **21**, **22** and **24**), described by simultaneous attractive/repulsive 3-D QSAR maps.

The effect of bulky substituents on the -**NH_2_** was further analyzed, starting from the tertiary nitrogen of **SCs** found in pyrrolidine, as in **9**–**16** (all IAs/FAs except **16**, Figure 6 and Appendix A), which adopted three different conformations. Favorably attracted by Val156 (as in **9**, **10**, and **12**; the positive STE OH2*_PLS-coefficients_* contours as evidence), the tertiary nitrogen provided good potency through moderate electrostatic interactions with Asp155 (distance over 6 Å, positive ELE OH2*_PLS-coefficients_* maps as validation). On the other hand, the interactions with Phe339 (observed for **11** and **16**) moved the heterocycle away from Asp155 into a negative STE OH2*_PLS-coefficients_* cloud, implying a need to maximally reduce bulkiness. Displacement of the heterocycle from Val156 towards Ser242 and Phe340, as in **13**–**15**, contributed more negatively than positively (a larger proportion of negative STE OH2*_PLS-coefficients_* maps than positive ones observed) to the potency, with a lack of interactions with Asp155.

#### 2.4.5. Generated 3-D QSAR Models’ Predictive Abilities

The Vina-predicted conformations and OH2 probe-derived BOTH 3-D QSAR models were evaluated for predictivity on TS compounds (Table 3 and Appendix A; Figure 7, Figure 8 and Appendix A).

The binding modes for all compounds except **25** (co-crystalized within the **7VOE** protein [28]) were modeled for **SCs** and were associated with experimentally available p*K*_i_ values. The p*K*_i_ values of TSs were predicted with an average absolute error of prediction (AAEPs) of 0.61 and 1.29 with the LOO and LSO CVs optimized models, respectively (Appendix A), to which the associated predictive *q^2^* (*q^2^*_pred_) values were calculated to be 0.43 and 0.307, respectively, confirming the good predictive capabilities of the model [79]. Similar to the **SCs**, ten TS compounds, namely **26**–**28**, **30**, **32**–**34**, **36**, **39**, and **41** (Figure 8 and Appendix A), filled only the hydrophobic pocket of the 5-HT_2A_R’s active site and were predicted in potency with the absolute error of prediction (AEP) below 1. Among them, the best predicted one was **33**, with errors of prediction (EPs) after LOO and LSO CVs of 0.03 and 0.16 p*K*_i_ units. The AEP was only 0.095 p*K*_i_ units, while the worst was **32**, associated with LOO and LSO CVs AEPs of 0.37 and 0.97 p*K*_i_ units, respectively. For the bipartite 5-HT_2A_R ligands, the AEP was greater than 1. The remaining compounds that were SB aligned throughout both the orthosteric region and the hydrophobic pocket of the receptor’s bipartisan active site. Within the subset, only compound **37** (Appendix A) potency was predicted with tolerable accuracy (LOO and LSO CVs AEPs of 1.2 and 0.94 p*K*_i_ units, respectively), whereas the worst predicted compound was **29** (Appendix A) (LOO and LSO CVs AEPs of 1.83 and 2.24 p*K*_i_ units, respectively).

Previously published CoMFA/CoMSIA LB 3-D QSAR models based on dibenzazecines [80], 3-aminoethyl-1-tetralones, piperazines, benzothiazepines, pyrrolobenzazepines [81], and arylpiperazines [82], as well as the LB GRID/GOLPE models of (aminoalkyl)benzo and heterocycloalkanones [83], were also more accurate in predicting the potencies of compounds occupying the hydrophobic pocket. On the other hand, within the molecular docking-based SB GRID/GOLPE 3-D QSAR models generated on either butyrophenones [84], or lozapine, ziprasidone, and ChEMBL-listed analogues [84], the quality of the alignment within either the orthosteric area, the hydrophobic pocket, or both, was, as here, evaluated using the highest *q*^2^ [63,85]. However, the universal SB 3-D QSAR model(s) defining the agonism/antagonism on the entire 5-HT_2A_R active site remains to be generated, perhaps after increasing the number of K*_i_*-associated co-crystallized 5-HT_2A_R compounds to a minimum of 15 (and thus updating Table 1), using either Open3DQSAR [77], 3-D_QSAutogrid/R [78], or Py_CoMFA [31,86].

### 2.5. Molecular Determinants for **SCs**

The pharmacodynamic profile obtained from the 3-D QSAR map analysis indicated some pharmacophoric features that may positively or negatively affect the potency of **SCs** as 5-HT_2A_R ligands. Therefore, comprehensive 3-D structure–activity relationship (SAR) rules were derived for the **SCs** structure–activity relationship model [67] and are shown in Figure 1 as a template (Figure 9). This led to the derivation of a unique SAR as a tool that could be used to drive the virtual screening campaign of new cathinones and similar compounds as 5-HT_2A_R ligands that could, as AGs, exert psychostimulant properties.

Thus, the central benzene ring (**Ph**) of **SCs** may be either unsubstituted or o-substituted proximal to the β-carbonyl group (**β-CO**) with bulkier portions to sterically engage Ser159, Thr160, and Ser242, while the o-bulkiness increase towards Trp336 and Phe340 would not be tolerated and HDB portions should be forced. As for the *m*-position on the **β-CO** side towards Ile163 and Trp164, **Ph** should either remain unsubstituted or be heavily substituted by a highly electronegative electron withdrawing/HBA group, while Trp336 on the opposite side would tolerate bulkier moieties with an HBD character. The p-position of **Ph** before Phe243, Phe332, Val333, and Phe340 could accommodate moieties, providing minimal van der Waals contacts with or substituted by a highly electronegative electron withdrawing/HBA group.

The **β-CO** pharmacophoric feature is responsible for the formation of HBs with Thr160, mitigated by the presence of either *p*-electronegative groups/HBA or bulky substituents (or functional groups containing each of these features) at the same position. Nevertheless, the most electronegative functional groups incorporated at the *p*-position of **Ph** may prevent the **β-CO**----HO-Thr160 HB formation. On the other hand, the incorporation of *m*- or *o*-electronegative groups/HBA (or corresponding hybrid functional groups) leads to the formation of **β-CO**----HO-Ser159 HB. Since none of the **SCs** formed the HB with Asp155 (an interaction that leads to the hallucinogenic effect of LSD [31]), the **β-CO** or similar **β-HBA** bioisosteric scaffold remains preferred for **SCs** to ensure the H-bonding described above, since the absence of **β-CO** resulted in the lowest potencies.

The **-CH_2_-** fragment substituted with a methyl group, located between the **β-CO** (or **Bn**) and -**NH_2_** (or -**NH**CH_3_), seemed to be the best choice to establish attractive steric interactions with Val156. The substitution of -**NH_2_** with bulkier acyclic and cyclic aliphatic moieties could also direct the **-CH_2_-**CH_3_ moiety to favorable van der Waals interactions with Phe339 and Phe340. The *m*-HBA/*o*-HBA-**Ph-β-CO**····HO-Ser159 HB architecture also supported the alignment of **-CH_2_-**CH_3_ with Phe339 and Phe340 but could also direct the moiety to unfavorable steric hindrance with Ser242. The **-CH_2_-**CH_3_ moiety should not be properly lengthened or branched, as the further increase of van der Waals interactions towards the listed residues could lead to decreased potency. Either the *m*-HBA-**Ph-β-CO**----HO-Thr160 HB or *p*-Me-**Ph-β-CO**····HO-Thr160 construct could force the **-CH_2_-**CH_3_ toward more unfavorable steric interactions with Trp336, a residue that would be more tolerant of the **-CH_2_-**HBD/HBA fraction.

The -**NH_2_** of the **SCs** should remain unsubstituted to maintain a salt bridge interaction with Asp155 and ensure high potency. Substitution with more voluminous moieties such as *N*-methyl/*N*-ethyl could be tolerated but could also result in penalizing van der Waals interactions with Leu228, Val336, Phe339, Trp336, and Tyr370, although not interfering with the -*N*-H moiety to form the required salt bridge. Finally, the tertiary nitrogen could provide attractive steric interactions with Val156 and interfere with Asp155, again resulting in lower potency.

### 2.6. External Validation of 3-D QSAR Models on Experimentally Determined 5-HT_2A_R Ligands

Finally, the suitability of the obtained 3D QSAR model to be used as a tool for virtual screening and potency prediction was evaluated on TS_CRY_ (Table 1, Figure 10). Since the bioactive conformations of the TS_CRY_ compounds were experimentally resolved, a deeper discussion in terms of their potency prediction was included concerning the TS compounds. Thus, the alignments of TS_CRY_ compounds were not as conserved as those of TS compounds, where most of the main cores of TS compounds were aligned with **SCs**, and for some of the compounds, the remaining parts occupied the orthosteric region. Therefore, the AAEPs of TS_CRY_s’ p*K*_i_ values were significantly higher for both LOO and LSO CVs optimized models, 1.12 and 1.49 (Appendix A), associated with *q^2^*_pred_ of 0.33 and 0.28, respectively, confirming the above-described 3-D QSAR model as a useful tool [79] to discover potential new chemical entities as 5-HT2AR ligands to be studied as broad psychedelic agents.

Thus, the best-predicted activities were for the **6A94** (Figure 11a,b), **6WH4** (Figure 11c,d), and **6WHA** (Figure 11e,f) ligands that were found deeply buried in the hydrophobic pocket, with AEPs of only 0.72, 0.95, and 1.05 p*K*_i_ units after the LOO CV, and 0.86, 1.16, and 1.69 p*K*_i_ units after the LSO CV, respectively. Moreover, given that the **6WH4** crystal is one of the most potent 5-HT_2A_R ligands known to date (Table 1), predicting its potency with acceptable error was very important for the present 3-D QSAR model, proving the model as capable of recognizing the highly potent ligands bound in a hydrophobic pocket. The main cores of **6WGT** (Appendix A), **7WC6** (Appendix A), and **7WC7** (Appendix A) were positioned at the top of the hydrophobic pocket, orthogonal to the SCs, and were within the scope of the 3-D QSAR *PLS-coefficients*, this time resulting in the LOO CV AEPs of 1.11 (for LSD) and 1.17 (for lisuride) p*K*_i_ units and LSO CV AEPs of 1.51 (for LSD) and 1.46 (for lisuride) p*K*_i_ units, respectively.

On the other hand, **6A93** (Appendix A), **7VOD** (Appendix A), and **7VOE** (Appendix A) crystals occupied both the orthosteric and hydrophobic pockets. The main core of **6A93** was superimposed by **SCs** and thus was the only part of the molecule covered by 3-D QSAR PLS*-coefficients*, as the rest filled the orthosteric cavity, for which the AEPs according to LOO and LSO CVs were 1.80 and 1.47 p*K*_i_ units, respectively. In contrast, **7VOD** and **7VOE** crystals were positioned above the **SCs** and only the central molecular regions were included in the 3-D QSAR PLS*-coefficients*, resulting in AEPs of 0.76 and 1.65 p*K*_i_ units after LOO cross-validation and 1.88 and 1.79 p*K*_i_ units after LOO cross-validation.

## 3. Materials and Methods

### 3.1. Crystal Structures Preparation

All selected 5-HT_2A_R and 5-HT_2B_R complexes (Table 1) available from the Protein Data Bank (https://www.rcsb.org/, accessed on 1 December 2022) were loaded into the UCSF Chimera v1.10.1 software [87] and were visually inspected. Complexes were superimposed using **5TVN** as a template (the best-resolved complex with a resolution of 1.6 Å) using the MatchMaker module and were then separated into chains using the command line implementation of the Chimera split command. Compared to other chains, chains A were complete with respect to the presence of antagonists and were retained. The antagonists were extracted from each chain A complex, completed by adding hydrogens appropriate for pH 7.4, and the AMBER parameters were calculated by Antechamber using a semi-empirical QM method. The protein parts of the stored monomers were improved by adding hydrogen atoms using the embedded leap module of the Amber 12 suite [88], after which the correct hydrogen atoms, appropriate for pH 7.4, were assigned to each amino acid residue. After preparation, the proteins were fused with the appropriate ligands and the complexes were energy minimized as follows. They were solvated by the Leap module with water molecules (TIP3P model, SOLVATEOCT Chimera command) in a box extending 10 Å in all directions, neutralized with either Na+ or Cl− ions, and refined by a single point minimization using the Sander module of the Amber suite with a maximum of 1000 steps of steepest descent energy minimization and a maximum of 4000 steps of conjugate gradient energy minimization, with an unbound cutoff of 5 Å. Minimized complexes were re-aligned (**5TVN** as a template), after which all ligands were extracted to compose the SB-aligned TR, ready to be used for the subsequent 3-D QSAR model building.

### 3.2. Alignment Assessment Rules

Structure-based alignment. Regarding SB alignment, a number of docking programs with all available scoring functions, free or open source for academic use, were evaluated to select the best one to reproduce the binding mode of 5-HT_2A_R and 5-HT_2B_R ligands. Namely, the programs were AutoDock [68], AutoDock Vina (hereafter referred to as Vina) [69], SMINA [71], DOCK [70], and PLANTS [72]. The entire SB procedure was evaluated using four levels of difficulty:Experimental Conformation Re-Docking (ECRD): a procedure in which the experimental conformations (EC) are flexibly docked back into the corresponding protein, evaluating the program for its ability to reproduce the observed bound conformations.Randomized Conformation Re-Docking (RCRD): a similar assessment to ECRD with the difference that the active site of protein is virtually occupied by conformations initially obtained from computational random optimization of corresponding co-crystallized molecules coordinates and positions. Thus, *ligands* were initially displaced from the active site and their experimental coordinates were changed by means of assigning new coordinates values: X = 0.000, Y = 0.000, Z = 0.000. Following that, allocated conformations were energy-minimized. Here the programs are evaluated for their ability to find the experimental pose, starting from the randomized minimized conformation.Experimental Conformation Cross-Docking (ECCD): comparable to ECRD, but the molecular docking was performed on all the TR proteins except the corresponding natives. Here the programs are evaluated to find the ligand binding mode in the active site such as the native one by means of amino acid configuration but are different in terms of amino acids induced-fit conformations, mimicking discrete protein flexibility at the same time.Randomized Conformation Cross-Docking (RCCD): same as the ECCD but using RCs as starting docking conformations. This is the highest level of difficulty since the program is demanded to dock any given molecule into an ensemble of protein conformations not containing the native one. The outcome is considered as the most important ability of the docking program, as the most accurate scoring function in the RCCD experiment is subsequently applied to any TS molecules whose experimental binding mode is unknown. The related docking accuracy (DA) is a direct function of the program’s probability to find a correct binding mode for an active molecule.

The alignment fitness was quantified through the evaluation of RMSD values and the subsequent docking accuracy (DA) values. As previously reported [31], DA can be used to test how the docking or alignment algorithms, respectively, are capable of predicting a ligand pose as close as possible to the experimentally observed and can be calculated by the following equation:*xA* = *f_rmsd_* ≤ *a* + 0.5 (*f_rmsd_* ≤ *b* − *f_rmsd_* ≤ *a*)(1)

In particular, *xA* is equal to DA in the case of docking accuracy, whereas *f_rmsd_* ≤ *a* and *f_rmsd_* ≤ *b* represent the fraction of aligned ligands, showing an RMSD value less than 2 Å or equal to 2 Å (*a* coefficient) and less than 3 Å or equal to 3Å (*b* coefficient), respectively. The widely accepted standard is that the correctly docked/aligned conformations are those displaying an RMSD value lower than 2 Å on all heavy atoms from the crystallographic structure of the ligand conformation, as found in the inhibitor–enzyme complex. Structures with RMSD between 2 and 3 Å are considered partially docked/aligned, whereas those with a RMSD higher than 3 were mis-docked and were thus not considered in the DA calculation.

#### 3.2.1. AutoDock Settings

For all ligands, the rigid root and rotatable bonds were defined using AutoDockTools. The docking was performed with AutoDock 4.2 by applying the cuboid docking grid coordinates, provided as follows: the xyz coordinates (in Ångströms) for the computation were Xmin/Xmax = −1.805/30.447, Ymin/Ymax = −12.113/9.482, Zmin/Zmax = 48.159/60.462. The coordinate setup was performed in a manner to embrace the minimized inhibitor, spanning 10 Å in all three dimensions. The Lamarckian Genetic Algorithm was used to generate orientations or conformations of ligands within the binding site. The procedure of the global optimization started with a sample of 200 randomly positioned individuals, a maximum of 1.0 × 10^6^ energy evaluations, and a maximum of 27,000 generations. A total of 100 runs were performed with RMS Cluster Tolerance of 0.5 Å.

#### 3.2.2. Vina Settings

The docking simulations were carried out with an energy range of 10 kcal/mol and exhaustiveness of 100 with RMS Cluster Tolerance of 0.5 Å, using the identical grid as for AutoDock4.2. The output comprised 20 different conformations for every receptor considered.

#### 3.2.3. Smina Settings

As Smina is AutoDock Vina fork, an identical setup was used as for Vina for either vina [69], vinardo [75], or ad4 [68] scoring functions.

#### 3.2.4. DOCK Settings

During the docking simulations with the DOCK program, the proteins were considered to be rigid while the inhibitors were regarded as flexible and were subjected to energy minimization. The solvent-accessible surface of each enzyme without hydrogen atoms was calculated using the DMS program [89], using a probe radius of 1.4 Å. The orientation of ligands was described using the SPHGEN and SPHERE_SELECTOR modules. A box around the binding site was constructed with the accessory module SHOWBOX. The steric and electrostatic environment of the pocket was evaluated with the program Grid using a 0.3 Å of grid spacing. Selected spheres were within 8 Å from ligand heavy atoms of the crystal structure, and for computing, the energy grids an 8 Å box margin and 6–9 VDW exponents were used.

#### 3.2.5. PLANTS Settings

The docking site was limited inside a 12 Å radius sphere, centered in the mass center of the crystallized ligand. Docking was performed by default settings using three different scoring functions: chemplp, plp, and plp95.

### 3.3. Generation of the TR and TS Designed Compounds

TR, TS, and designed compounds were modelled by applying the Chemaxon’s msketch module [90] through molecular mechanic optimization, upon which the hydrogen atoms appropriate to pH 7.4 were assigned. Upon the generation of structures, compounds were uploaded into previously described SB to obtain the bioactive conformations.

### 3.4. Retrieval of 5-HT_2A_R-Lipid Bilayer Complex

The crystal structure of 5-hydroxytryptamine receptor 2A in complex with risperidone, deposited at PDB under the code: **6A93**, was found modeled in dipalmitoylphosphatidylcholine (DPPC) bilayer system and deposited as entry: 4282 at Orientations of Proteins in Membranes (OPM) database (https://opm.phar.umich.edu/, accessed on 2 December 2022). Upon retrieval, the 5-HT_2A_R-DPPC complex was subjected to the same preparation protocol with UCSF Chimera, as described in Section 3.1.

### 3.5. Structure Alignment of LSD, TR, TS, and Designed Compounds within 5-HT_2A_R

The same relative coordinates were assigned to the 5-HT_2A_R- DPPC system as for 5-HT_2B_R, using the UCSF Chimera’s MatchMaker module. Afterwards, either LSD, TR, TS, or designed compounds were docked into 5-HT_2A_R-DPPC complex using the best performing algorithm/scoring function.

### 3.6. 3-D QSAR Models Generation

The herein Vina-based TR was submitted to Open3DQSAR procedure to generate partial least squares (PLS)-based 3-D QSAR models using eight probes, namely CR (alkyl carbon), CB (aromatic carbon), NC=O (amide nitrogen), NR (amine nitrogen), O=C (carbonyl oxygen), OH2 (oxygen in water), HNCO (amide hydrogen), and ELE (electrostatic field). The 3D QSAR models were built for each probe using a maximum of 5 principal components. Initially generated molecular interaction fields (MIFs) and PLS 3-D QSAR models were first optimized on 1 Å grid spacing using the standard pretreatment protocol (energy cutoff of ± 5 Kcal/mol, zeroing = 0.01 Kcal/mol, and minimum standard deviation = 0.05) and by storing the corresponding standard (*r*^2^) and cross-validated (*q*^2^) correlation coefficients. A second stage of optimization was achieved by a Variable Pretreatment Optimization (VPO) procedure using Leave-One-Out (LOO) and Leave-Some-Out (LSO) cross-validation while monitoring *q*^2^. Full pretreatment of the data derived from the MIFs calculations was performed by exploring the combinations of cutoff values from −5 to 5 kcal/mol with intervals between the cutoffs equal to 1, zero values from −0.005 to 0.05 kcal/mol with an interval of 0.005, and standard deviation values from −0.01 to 0.1 with an interval of 0.01. The final 3-D QSAR models (Table 7) were obtained by the fractional factorial design (FFD) feature selection procedure. The internal validation (robustness) of the 3-D QSAR models was evaluated by classical cross-validation (CV) techniques (LOO and LSO), while the lack of random correlation was evaluated by CV and Y-scrambling combination (Table 7). The best 3D QSAR models were used to predict the activity of either co-crystalized TR compounds (Table 1) or SB-aligned TS compounds (Table 3).

## 4. Conclusions

The enclosed manuscript summarizes the efforts to define the pharmacodynamics of **SCs** available in the literature [21,34] as 5-HT_2A_R ligands, namely agonists (i.e., psychedelic abusers) and IAs/PAs (i.e., non-psyshostimulants). The bioactive conformations of **SCs** were obtained using the AutoDock Vina [69] after exhaustive SB alignment evaluation on co-crystallized 5-HT_2A_R and 5-HT_2B_R ligands, while the crucial interactions leading to the inhibition constants and possible psychedelic effects for AGs were described by a statistically robust CoMFA-like 3-D QSAR model, as generated by Open3DQSAR [77], starting from the OH2 probe, using both steric and electrostatic fields. The obtained *PLS-coefficients* maps were summarized in a 3-D SAR model (i.e., molecular determinants), a useful tool for virtual screening campaigns in the search for new SCs with potential psychedelic effects with an etiology in the agonism of 5-HT_2A_R. The applicability of the OH2 probe 3-D QSAR model as a predictive engine was evaluated on co-crystallized 5-HT_2A_R ligands, showing satisfactory predictive properties, considering that none of the experimentally resolved 5-HT_2A_R ligands were structural homologs of **SCs**.

## Figures and Tables

**Figure 1 molecules-28-06236-f001:**
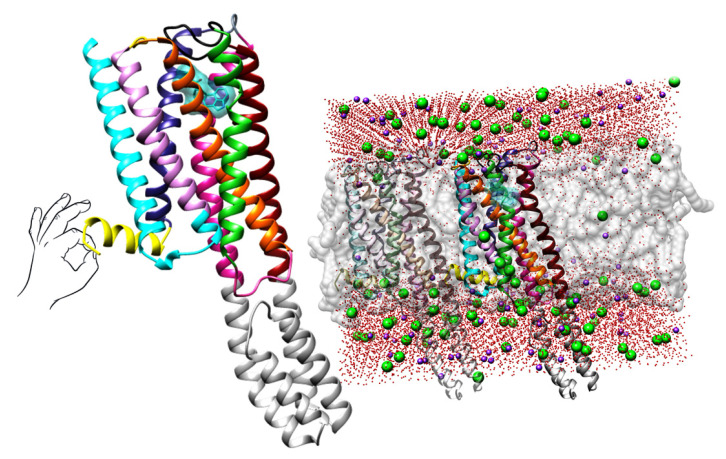
Left: The crystal structure of 5-HT_2A_R in complex with LSD (as deposited at PDB, https://www.rcsb.org/, PDB ID: **6HWT** [16], accessed on 1 December 2022). LSD is depicted in blue encircled by the transparent sphere, TM1 (residues 69–101) is colored in blue, TM2 (residues 111–137) is depicted in plum, TM3 (residues 114–179) is painted in orange, TM4 (residues 188–217) is colored in lime green, TM5 (residues 231–265) is depicted in dark red, TM6 (residues 315–349) is painted in violet-red, TM7 (residues 353–383) is colored in dark grey, intracellular amphipathic helix H8 (residues 384–399) is depicted in yellow, two intracellular loops ICL1 and ICL2 (residues 102–111 and 179–187) are illustrated in cyan and hot pink, respectively, three extracellular loops ECL1, ECL2, and ECL3 (residues 134–138, 218–230, and 350–352) are labeled in gold, dim gray, and dark slate gray, respectively; Right: The crystal structure of 5-HT_2A_R immersed into dipalmitoylphosphatidylcholine (DPPC) lipid bilayer (as deposited at OPM, https://opm.phar.umich.edu/, entry: 4282, accessed on 2 March 2023): the DPPCs are illustrated with transparent gray spheres, oxygen atoms from water molecules are colored in red, sodium atoms are presented as pink spheres, and chlorine atoms are described as green spheres. For the clarity of the presentation, hydrogen atoms were omitted.

**Figure 2 molecules-28-06236-f002:**
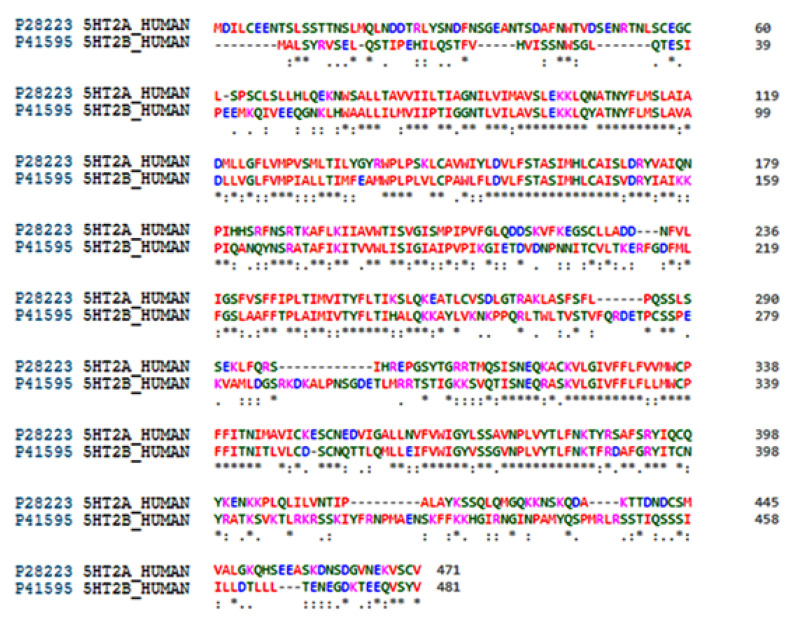
Sequence alignment between human 5-HT_2A_R (Uniprot entry P28223) and 5-HT_2B_R (UniProt entry P41596) performed by Clustal Omega [66]. Special characters meanings: * (Asterix) positions with a single, fully conserved residue; “:” (colon) positions with conservation between amino acid groups of similar properties; “.” (period) positions with conservation between amino acid groups of weakly similar properties.

**Figure 3 molecules-28-06236-f003:**
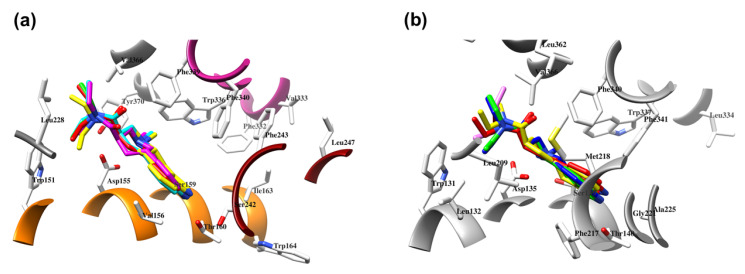
The docking assessment of LSD co-crystallized into 5-HT_2A_R (the **6WGT** crystal [16]), EC conformation in pink, ECRD conformation in blue, RCRD conformation in green, ECCD conformation in red, and RCCD conformation in yellow (**a**); LSD co-crystallized into 5-HT_2B_R (the **5VNT** crystal [31]), EC conformation in pink, ECRD conformation in blue, RCRD conformation in green, ECCD conformation in red, and RCCD conformation in yellow (**b**).

**Figure 4 molecules-28-06236-f004:**
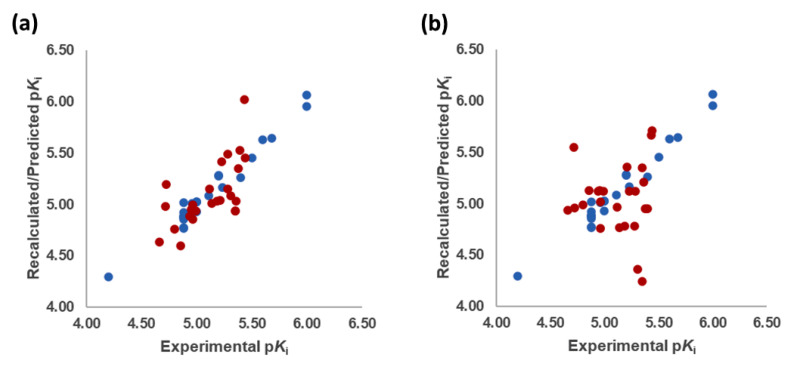
Experimental versus recalculated (blue circles) and predicted (red circles) pKis for TR using OH2 probe BOTH (STE + ELE) LOO model at 5 PCs (**a**); OH2 probe BOTH (STE + ELE) LSO model at 5 PCs (**b**).

**Figure 5 molecules-28-06236-f005:**
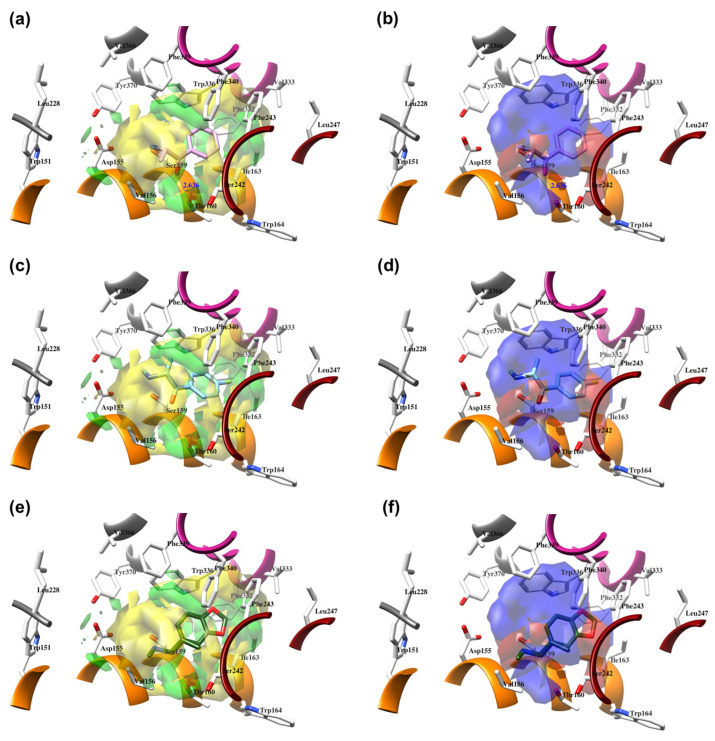
The structure-based alignment within 5-HT_2A_R-DPPC and OH2 probe-based *PLS-coefficients* (positive steric coefficients presented in green, negative steric coefficients depicted in yellow, positive electrostatic coefficients portrayed in red, negative electrostatic coefficients displayed in blue) for **1** (**a**,**b**); **2** (**c**,**d**); **17** (**e**,**f**). For the clarity of presentation, DPPC was omitted.

**Figure 6 molecules-28-06236-f006:**
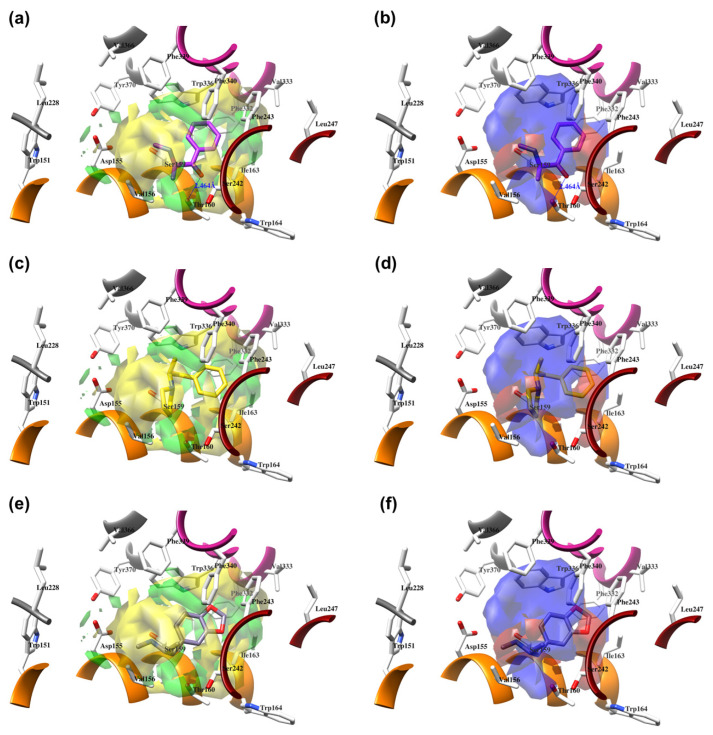
The structure-based alignment within 5-HT_2A_R-DPPC and OH2 probe-based *PLS-coefficients* (positive steric coefficients presented in green, negative steric coefficients depicted in yellow, positive electrostatic coefficients portrayed in red, negative electrostatic coefficients displayed in blue) for **4** (**a**,**b**), **9** (**c**,**d**), and **18** (**e**,**f**). For the clarity of presentation, DPPC was omitted.

**Figure 7 molecules-28-06236-f007:**
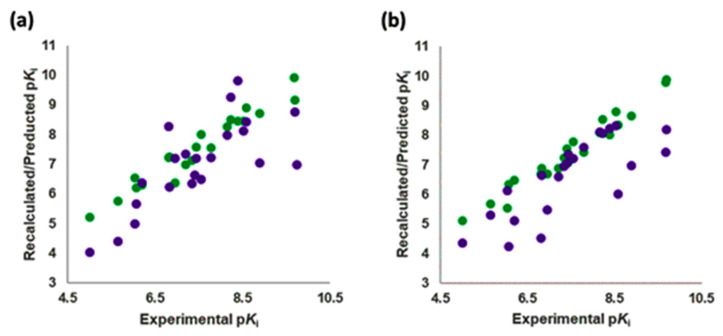
Experimental versus recalculated (green circles) and predicted (purple circles) p*K*_i_s for TS using OH2 probe BOTH (STE + ELE) LOO model at 5 PCs (**a**); OH2 probe BOTH (STE + ELE) LSO model at 5 PCs (**b**).

**Figure 8 molecules-28-06236-f008:**
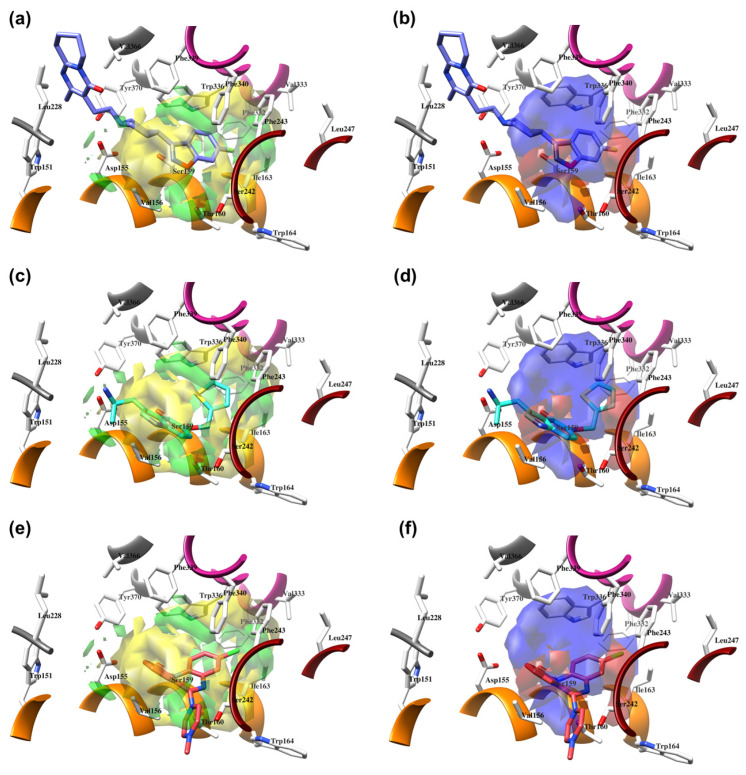
The structure-based alignment within 5-HT_2A_R-DPPC and OH2 probe-based *PLS-coefficients* (positive steric coefficients presented in green, negative steric coefficients depicted in yellow, positive electrostatic coefficients portrayed in red, negative electrostatic coefficients displayed in blue) for **25** (**a**,**b**); **26** (**c**,**d**); **27** (**e**,**f**). For the clarity of presentation, DPPC was omitted.

**Figure 9 molecules-28-06236-f009:**
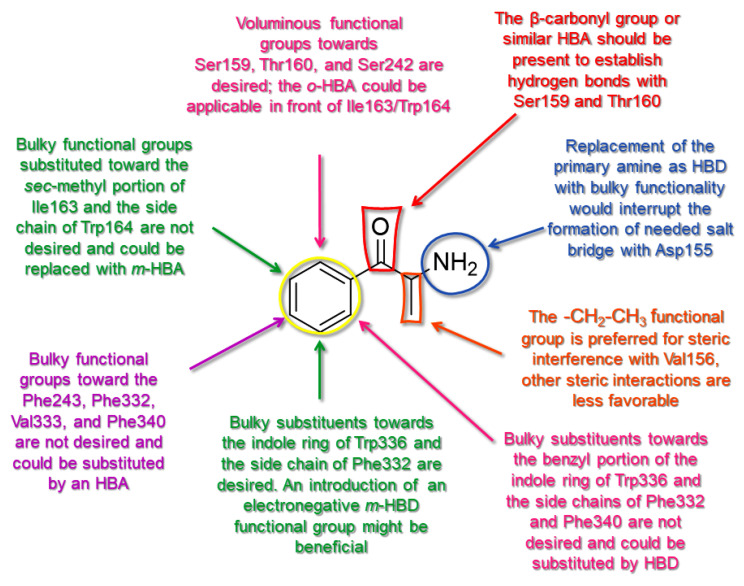
The SAR model for **CSs** as 5-HT_2A_R ligands and agonists is seen through **1**.

**Figure 10 molecules-28-06236-f010:**
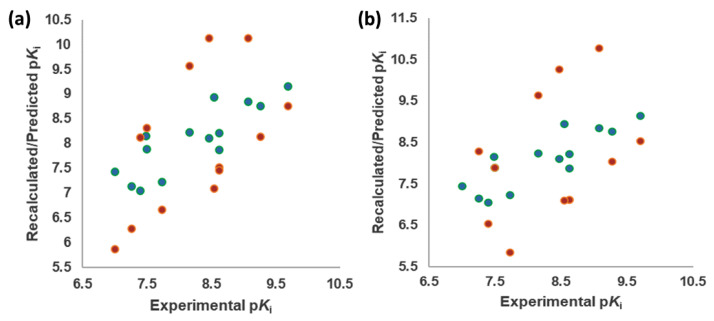
Experimental versus recalculated (green circles) and predicted (orange circles) p*K*_i_s for TS_CRY_ using OH2 probe BOTH (STE + ELE) LOO model at 5 PCs (**a**); OH2 probe BOTH (STE + ELE) LSO model at 5 PCs (**b**).

**Figure 11 molecules-28-06236-f011:**
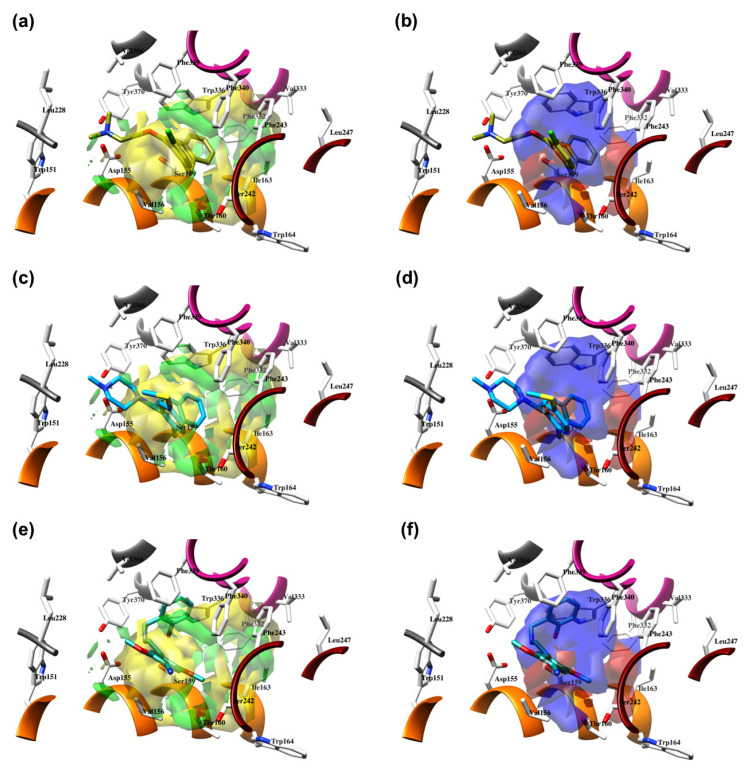
The structure-based alignment within 5-HT_2A_R-DPPC and OH2 probe-based *PLS-coefficients* (positive steric coefficients presented in green, negative steric coefficients depicted in yellow, positive electrostatic coefficients portrayed in red, negative electrostatic coefficients displayed in blue) for **6A94** (**a**,**b**); **6WH4** (**c**,**d**); **6WHA** (**e**,**f**). For the clarity of presentation, DPPC was omitted.

**Table 4 molecules-28-06236-t004:** The RMSD matrix of either 5-HT_2A_R or 5-HT_2B_R proteins bound to LSD.

	5TVN	6DRX	6WGT	7SRR	7WC6
**5TVN**	0.000	1.012	0.731	1.154	1.077
**6DRX**	1.012	0.000	0.795	1.135	1.165
**6WGT**	0.731	0.795	0.000	0.765	0.727
**7SRR**	1.154	1.135	0.765	0.000	1.195
**7WC6**	1.077	1.165	0.765	1.195	0.000

**Table 5 molecules-28-06236-t005:** Structure-based alignment assessment of 5-HT_2A_R ligands within the RCCD stage.

Code	AutoDock	Vina	SMINA			DOCK	PLANTS			Ref.
SF ^a^			Vina	Vinardo	ad4		chemplp	plp	plp95	
	Randomized Conformation Cross-Docking (RCCD)									
**6A93 ^b^**	2.784 ^c^	2.843	2.742	2.563	3.563	3.231	2.573	2.742	2.746	[16]
**6A94**	2.657	2.224	3.241	3.431	3.431	3.422	2.991	2.664	2.561	[16]
**6WGT**	2.943	1.943	2.941	2.567	2.567	2.993	2.892	3.245	2.993	[16]
**6WH4**	2.995	1.971	2.518	2.783	2.783	2.961	2.973	2.835	3.762	[25]
**6WHA**	3.726	2.941	3.663	3.452	3.452	2.693	2.116	2.985	3.426	[25]
**7RAN**	3.651	2.365	2.632	2.954	3.954	3.652	2.639	2.954	3.624	[27]
**7VOD**	3.648	1.984	3.654	3.642	3.642	2.584	2.652	2.667	3.621	[28]
**7VOE**	6.257	1.965	2.548	2.984	2.984	2.524	2.547	2.647	2.457	[28]
**7WC4**	3.658	2.695	2.458	2.658	2.658	3.632	3.698	3.965	3.621	[26]
**7WC5**	2.398	2.657	2.654	2.984	2.984	3.324	3.258	3.541	2.514	[26]
**7WC6**	3.695	3.625	3.254	3.652	3.652	2.659	2.987	2.564	2.558	[26]
**7WC7**	2.698	3.954	2.698	2.874	2.874	2.584	2.898	2.842	2.774	[26]
**7WC8**	3.654	3.625	2.548	2.636	2.636	3.395	2.235	2.665	2.664	[26]
**7WC9**	3.658	3.695	2.547	2.584	2.584	2.987	2.397	2.635	2.981	[26]
**DA ^d^**	21.43%	50.00%	35.71%	35.71%	28.57%	28.57%	42.85%	42.85%	32.14%	

^a^ Scoring function; ^b^ Experimental conformation; ^c^ Root-Mean-Square-Deviation measured between the heavy atoms of the ligand’s experimental and the ligand’s re-/cross-aligned conformation; ^d^ Docking accuracy.

**Table 6 molecules-28-06236-t006:** Structure-based alignment assessment of 5-HT_2B_R ligands.

Code	AutoDock	Vina	SMINA			DOCK	PLANTS			Ref.
SF ^a^			Vina	Vinardo	ad4		chemplp	plp	plp95	
	Randomized Conformation Cross-Docking (RCCD)									
**4NC3 ^b^**	4.454 ^c^	2.785	2.986	2.784	2.984	4.313	3.125	2.742	3.421	[29]
**4IB4**	3.431	2.895	4.124	3.984	2.634	3.453	3.784	2.563	2.741	[29]
**5TUD**	3.625	2.748	2.642	3.614	3.569	3.695	2.539	2.597	3.625	[57]
**5TVN**	2.895	1.993	1.963	1.943	3.964	3.241	2.431	3.254	2.346	[31]
**6DRX**	2.674	2.864	2.424	2.435	2.874	3.254	2.964	3.541	2.758	[33]
**6DRY**	2.895	2.998	3.894	3.784	2.695	2.451	2.531	2.431	2.321	[33]
**6DRZ**	2.992	2.674	3.431	3.895	3.624	2.563	2.728	3.522	2.462	[33]
**6DS0**	2.567	1.864	2.974	2.983	2.874	2.451	2.351	2.214	3.325	[33]
**7SQR**	2.517	2.957	2.987	2.524	2.595	3.648	2.487	2.845	2.635	[32]
**7SRS**	2.698	1.997	1.987	1.987	2.749	2.874	2.228	2.457	3.984	[32]
**7SRR**	3.624	2.487	2.964	1.987	2.996	3.625	2.487	2.625	3.635	[32]
**DA ^d^**	31.82%	63.64%	45.45%	45.45%	36.36%	18.18%	40.91%	36.36%	31.82%	

^a^ Scoring function; ^b^ Experimental conformation; ^c^ Root-Mean-Square-Deviation measured between the heavy atoms of the ligand’s experimental and the ligand’s re-/cross-aligned conformation; ^d^ Docking accuracy.

**Table 7 molecules-28-06236-t007:** Statistical results of the best Open3DQSAR models derived after VPO and FFD optimization.

Probe	Field	GS ^a^	PC ^b^	CO ^c^	Z ^d^	SD ^e^	*r* ^2 f^	*q*^2^_LOO_ ^g^	*q*^2^_LSO_ ^h^	*r*^2^_YS_ ^i^	*q*^2^_YS LOO_ ^j^	*q*^2^_YS LSO_ ^k^
CR	STE	1	5	4	0.05	0.01	0.932	0.552	0.476	0.864	−0.217	−0.264
	ELE	1	5	4	0.05	0.01	0.956	0.476	0.411	0.812	−0.231	−0.284
	BOTH	1	5	4	0.05	0.01	0.961	0.512	0.417	0.856	−0.231	−0.246
CB	STE	1	4	4	0.01	0.01	0.943	0.473	0.442	0.836	−0.251	−0.256
	ELE	1	4	4	0.01	0.01	0.967	0.414	0.412	0.822	−0.236	−0.238
	BOTH	1	4	4	0.01	0.01	0.934	0.486	0.446	0.821	−0.245	−0.217
NC=O	STE	1	5	3	0.02	0.04	0.951	0.536	0.498	0.861	−0.236	−0.284
	ELE	1	5	3	0.02	0.04	0.931	0.254	0.436	0.856	−0.258	−0.264
	BOTH	1	5	3	0.02	0.04	0.962	0.521	0.432	0.754	−0.312	−0.157
NR	STE	1	4	5	0.03	0.01	0.943	0.484	0.461	0.814	−0.264	−0.264
	ELE	1	4	5	0.03	0.01	0.945	0.512	0.421	0.806	−0.254	−0.235
	BOTH	1	4	5	0.03	0.01	0.952	0.504	0.412	0.825	−0.147	−0.135
O=C	STE	1	3	5	0.05	0.02	0.912	0.501	0.495	0.841	−0.264	−0.244
	ELE	1	3	5	0.05	0.02	0.925	0.482	0.472	0.822	−0.254	−0.251
	BOTH	1	3	5	0.05	0.02	0.976	0.498	0.426	0.734	−0.137	−0.146
OH2	STE	1	5	5	0.05	0.02	0.973	0.684	0.594	0.845	−0.264	−0.241
	ELE	1	5	5	0.05	0.02	0.981	0.562	0.534	0.824	−0.247	−0.224
	BOTH	1	5	5	0.05	0.02	0.971	0.671	0.316	0.842	−0.145	−0.321

^a^ Grid spacing (in Å); ^b^ Optimal number of principal components/latent variables; ^c^ CutOff values ± interval in kcal/mol; ^d^ Zeroing values in kcal/mol; ^e^ Standard deviation; ^f^ Conventional square-correlation coefficient; ^g^ Cross-validation correlation coefficient using the leave-one-out method; ^h^ Cross-validation correlation coefficient using the leave-some-out (LSO) method with 5 random groups; ^i^ Average square correlation coefficient obtained after Y-scrambling process. ^j^ Average cross-validation correlation coefficient using the leave-one-out (LOO) method obtained after Y-scrambling process. ^k^ Average cross-validation correlation coefficient using the leave-some-out (LSO) method with five random groups obtained after Y-scrambling process.

## Data Availability

Complexes herein used to derive 3-D QSAR models and perform SB alignments assessments are available at Protein Data Bank (https://www.rcsb.org/). Structures of other training set and test set compounds are retrieved from literature. The obtained 3-D QSAR models are available upon request from Milan Mladenović (E-mail: milan.mladenovic@pmf.kg.ac.rs).

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
