# Peer review of "Molecular Docking Assessment of Cathinones as 5-HT2AR Ligands: Developing of Predictive Structure-Based Bioactive Conformations and Three-Dimensional Structure-Activity Relationships Models for Future Recognition of Abuse Drugs"

_molecules, 2023, doi:10.3390/molecules28176236_

Round 1

Reviewer 1 Report

The manuscript "Molecular Docking Assessment of Cathinones as 5-HT2AR Stimulators: Developing of Predictive Structure-Based Bioactive Conformations and Three-Dimensional Structure-Activity Relationships Models for Future Recognition of Abuse Drugs" is a fairly thorough Computational study of the relationship between the 3D structure of SCs and their affinity to 5-HT2AR , which includes molecular docking and three-dimensional quantitative structure-activity relationship (3-D QSAR) studies, the results of which are interesting not only from the point of view of theoretical chemistry, but also in the future for drug design.

However, some points should be noted

 -It is highly recommended to extend tables 1-3 with data on the ligand mediated effects (activator, inhibitor, agonist, antagonist, etc.) This will facilitate the perception of an array of data on the action of ligands of various structures.

-the binding affinities are usually expressed as equilibrium dissociation constants (KD values (M)).

- Check the phrase "the presence of unsubstituted Ph, as in 1 (Figures 5a and 65) (line 440)" please.

-Since these compounds are multi-targeted and considering their pharmacodynamic properties, the correlation of their biological activity only with Kis with respect to 5-HT2aR does not seem to be fully justified.

- Within the framework of a scientific article, it is useful to use the terms agonist, antagonist, ligand accepted in scientific lexis instead of stimulator, binder, etc.   “5-HT2BR binders” - the term is completely unacceptable.

 - How critical for affinity/activity is the replacement "electrostatic interactions with TM3 Asp155, instead of a hydrogen bond (HB)",  line 318  . What is the reason for such significance of the type of interaction, data on the strength of these bonds are not given.

- The authors consider in detail the role of structural elements of SCs in binding to 5-HT2aR. It should be more precisely expressed what the authors mean by the term stimulators, if only the effect on binding affinity to a given receptor is evaluated, how the proposed models suggest predicting differently directed biological effects if any. This point needs to be discussed in detail.

- the desire of the authors to use a variety of vocabulary is very commendable, but in this case, within the framework of a scientific article, it is better to confine ourselves to generally accepted terminology: agonist, antagonist, ligand, etc.

Author Response

Reviewer 1

The manuscript "Molecular Docking Assessment of Cathinones as 5-HT2AR Stimulators: Developing of Predictive Structure-Based Bioactive Conformations and Three-Dimensional Structure-Activity Relationships Models for Future Recognition of Abuse Drugs" is a fairly thorough Computational study of the relationship between the 3D structure of SCs and their affinity to 5-HT2AR, which includes molecular docking and three-dimensional quantitative structure-activity relationship (3-D QSAR) studies, the results of which are interesting not only from the point of view of theoretical chemistry but also in the future for drug design.

Dear Sir/Madam,         

All your comments and remarks were carefully considered and below you can find the list of responses to each of the raised questions. We are confident that all your suggestions helped us significantly improve the quality of the manuscript.

However, some points should be noted

 -It is highly recommended to extend tables 1-3 with data on the ligand-mediated effects (activator, inhibitor, agonist, antagonist, etc.) This will facilitate the perception of an array of data on the action of ligands of various structures.

Thank you for this valuable remark. According to your suggestion, Tables 1-3 were completed with the required data.

-the binding affinities are usually expressed as equilibrium dissociation constants (KD values (M)).

Dear Sir/Madam,

Thank you for having pointed this issue. Throughout the text, the string “binding affinities” has been changed to “inhibition constants”.

- Check the phrase "the presence of unsubstituted Ph, as in 1 (Figures 5a and 65) (line 440)" please.

Dear Sir/Madam,

Thank you for this valuable remark. We went over the whole manuscript to improve the Scientific English.

-Since these compounds are multi-targeted and considering their pharmacodynamic properties, the correlation of their biological activity only with Kis with respect to 5-HT2aR does not seem to be fully justified.

Dear Sir/Madam,

Thank you for this valuable comment. We strongly agree with you that the correlation of cathinones’ biological activity only with Kis concerning 5-HT2AR does not seem to be fully justified. However, for only a few of herein discussed synthetic cathinones more pharmacologically relevant data in terms of drug abuse, such as the onset of action (OA) and duration of action (OA) is available (new Table S1 in Supplementary Materials). On the other hand, given that all the compounds were associated with Kis, regression models were done using them.

- Within the framework of a scientific article, it is useful to use the terms agonist, antagonist, and ligand accepted in scientific lexis instead of stimulator, binder, etc.   “5-HT2BR binders” - the term is completely unacceptable.

Thank you for this suggestion. Given that herein discussed synthetic cathinones display different pharmacology, in the Title of the paper, the word “stimulators” was replaced with the word “ligands”, whereas throughout the manuscript the phrase “5-HT2BR binders” was replaced with more pharmacologically accurate terms, like agonists, inverse agonists, partial agonists, and full antagonists, depending on context.

 - How critical for affinity/activity is the replacement "electrostatic interactions with TM3 Asp155, instead of a hydrogen bond (HB)”, line 318. What is the reason for such significance of the type of interaction? Data on the strength of these bonds are not given.

Thank you for this question. In literature, the psychedelic activity of LSD is reported to be directly associated with a hydrogen bond between its tertiary amine and TM3 Asp155. Consequently, the lower psychostimulant activity of synthetic cathinones compared to LSD can be associated with the fact that none of them was found to establish the hydrogen bond with TM3 Asp155.

- The authors consider in detail the role of structural elements of SCs in binding to 5-HT2aR. It should be more precisely expressed what the authors mean by the term stimulators, if only the effect on binding affinity to a given receptor is evaluated, and how the proposed models suggest predicting differently directed biological effects if any. This point needs to be discussed in detail.

Thank you for this point. Accordingly, we changed the term “stimulators” to agonists, inverse agonists, partial agonists, and full antagonists where needed.

As for the model’s predictions, from the literature reports it is very hard to conclude the exact pharmacology of SCs as ligands of 5-HT2aR and consequent psychedelic effects, as not all the compounds have been directly associated with agonism, partial/inverse agonism, or antagonism, to the best of our knowledge only 4-Bromomethcathinone, 3-Bromomethcathinone, 2-Fluoromethcathinone, 2-Trifluoromethoxy-methcathinone, α-PPP, 4-Methyl-α-PPP, 4-Bromo-α-PPP, 3-Bromo-α-PPP, Naphyrone, MDPV, Pyrovalerone, and MDPPP (which is now clearly indicated in the text). For other compounds, it is considered that those which possess some psychedelic effects are agonists and those which mitigate them/do not possess them are inverse agonists or antagonists. The exact way of thinking was applied here as well (also clearly indicated in the text).

With the enclosed models a focus was given more to describing the compounds binding modes and either positively or negatively contributing functional groups and then to associating them as much as possible with actual pharmacology and consequent psychedelic effects. In that sense, the models are, at least from our point of view, very applicable.

Comments on the Quality of English Language

- the desire of the authors to use a variety of vocabulary is very commendable, but in this case, within the framework of a scientific article, it is better to confine ourselves to generally accepted terminology: agonist, antagonist, ligand, etc.

Thank you for this valuable note. Pharmacologically accurate terms, like agonists, inverse agonists, partial agonists, and full antagonists, are now used where needed.

Reviewer 2 Report

predictive structure-based bioactive conformations can be used to recognize abuse drugs by modeling the bioactive conformations of cathinones using molecular docking and three-dimensional quantitative structure-activity relationship (3-D QSAR) studies. The resulting structure-based (SB) 3-D QSAR models can then be used to identify molecular determinants by which any untested cathinone molecule can be predicted as a potential 5-HT2AR binder prior to experimental evaluation. This approach could potentially be applied to other drugs as well, allowing for the identification of new compounds with similar pharmacodynamic profiles to known abuse drugs.

Author Response

Dear Sir/Madam,         

Thank you for these kind words referring to the quality of our manuscript.
